# Image-based identification and isolation of micronucleated cells to dissect cellular consequences

Lucian DiPeso[1,2], Sriram Pendyala[3], Heather Z Huang[1], Douglas M Fowler[3], Emily M Hatch[1,4]*

[1]Basic Sciences Division, Fred Hutchinson Cancer Center, Seattle, United States; [2]Molecular & Cellular Biology, University of Washington, Seattle, United States; [3]Genome Sciences, University of Washington, Seattle, United States; [4]Human Biology Division, Fred Hutchinson Cancer Center, Seattle, United States

## eLife Assessment

This **valuable** paper reports machine learning-based image analysis pipelines for the automated segmentation of micronuclei and the detection and sorting of micronuclei-containing cells. These are powerful new tools for researchers who study micronuclei and their physiologic consequences. The analysis of the new tools and their benchmarking is rigorous and **convincing**; applications and remaining limitations are well explained in the paper.

*For correspondence:
ehatch@fredhutch.org

Competing interest: The authors declare that no competing interests exist.

**Abstract** Recent advances in isolating cells based on visual phenotypes have transformed our ability to identify the mechanisms and consequences of complex traits. Micronucleus (MN) formation is a frequent outcome of genome instability, triggers extensive changes in genome structure and signaling coincident with MN rupture, and is almost exclusively defined by visual analysis. Automated MN detection in microscopy images has proved challenging, limiting discovery of the mechanisms and consequences of MN. In this study we describe two new MN segmentation modules: a rapid model for classifying micronucleated cells and their rupture status (VCS MN), and a robust model for accurate MN segmentation (MNFinder) from a broad range of cell lines. As proof-of-concept, we define the transcriptome of non-transformed human cells with intact or ruptured MN after chromosome missegregation by combining VCS MN with photoactivation-based cell isolation and RNASeq. Surprisingly, we find that neither MN formation nor rupture triggers a strong unique transcriptional response. Instead, transcriptional changes appear correlated with small increases in aneuploidy in these cell classes. Our MN segmentation modules overcome a significant challenge with reproducible MN quantification, and, joined with visual cell sorting, enable the application of powerful functional genomics assays to a wide-range of questions in MN biology.

## Introduction

Recent advances in automated image analysis have led to the development of high-throughput platforms to isolate specific cell classes and match visual phenotypes to specific genetic and expression profiles. These platforms bring the power of population-based analyses and pooled genetic screening to a large range of phenotypes that are defined solely by visual changes in subcellular features. One such feature are micronuclei (MN), nuclear compartments containing a few chromosomes or chromatin fragments that result from mitotic segregation errors or DNA damage (*Bona and Bakhoum, 2024*; *Guo et al., 2019*). Increased MN frequency is a hallmark of carcinogen exposure, cancer development,

**eLife digest** Healthy cells house most of their genetic information, such as their chromosomes, within a dedicated compartment known as the nucleus. Micronuclei, on the other hand, are small cellular compartments which contain pieces of DNA left behind after improper cell divisions or other abnormal events. Detectable under the microscope, their presence is usually interpreted as a sign of aging, disease or negative perturbations such as exposure to toxic chemicals.

However, new research suggests that micronuclei could also be directly harmful. When they rupture – which they almost always do – the sudden presence of unprotected genetic information in the 'incorrect' part of the cell could trigger dangerous cascades of events. To better understand these processes, researchers need to be able to quickly identify and isolate micronuclei-carrying cells within a large population, but such techniques are currently lacking.

DiPeso et al. aimed to address this gap by developing an AI tool, MNFinder, which can analyse images of live cells and automatically identify micronuclei. This allowed the team to mark and isolate cells in which these compartments were either absent, present, or had ruptured. Further experiments showed that, unexpectedly, patterns of gene expression did not differ between these different groups of cells. This suggests that the cell cannot 'see' that it has too many nuclei, or sense when its own DNA is in the wrong place. The disease-associated changes caused by the rupturing of micronuclei may therefore only emerge later, when the cell next divides. At this point, the DNA is heavily damaged and can activate immune checkpoint pathways, which can generate cancer-driving mutations and cellular self-destruct signals.

By developing and making MNFinder freely available, DiPeso et al. hope that micronuclei will be easier to study for scientists in a wide range of fields. Interestingly, micronuclei occur frequently during early human embryo development and may affect fertility, highlighting another research area that could benefit from these tools. As with all image analysis programs, however, increased usage, data, and model training will expand the use and ease of applying these tools to critical questions in human health.

and aging, and MN are potent drivers of massive genome structural changes, pro-inflammatory and metastasis signaling, and senescence (*Bakhoum et al., 2018*; *Dou et al., 2017*; *Harding et al., 2017*; *He et al., 2019*; *Mackenzie et al., 2017*; *Mohr et al., 2021*; *Soto et al., 2018*; *Zhang et al., 2015*). These processes are linked to MN rupture, which exposes the chromatin to the cytosol for the duration of interphase (*Hatch et al., 2013*), and may contribute to tumorigenesis, metastasis, aging, and immune disorders (*Bona and Bakhoum, 2024*; *Guo et al., 2019*).

Most studies on the biology and consequences of MN formation and rupture take advantage of the fact that MN can be induced with high frequency in cultured cells, for instance by inhibiting the spindle assembly checkpoint kinase Mps1 (*Krupina et al., 2021*). However, these interventions cause diverse additional nuclear and cellular changes, including nuclear atypia, chromatin bridges, aneuploidy, and DNA damage (*Flynn et al., 2021*; *Santaguida et al., 2017*). Sophisticated techniques have been developed to overcome this challenge by identifying micronucleated cells in single cell arrays, inducing missegregation of a single chromosome or chromosome arm without global ploidy changes by CRISPR, and purifying MN from lysed cells (*Agustinus et al., 2023*; *Ly et al., 2017*; *Mohr et al., 2021*; *Papathanasiou et al., 2023*; *Zhang et al., 2015*). These techniques led to new insights into MN functional defects and their mechanisms but have a limited ability to define the cellular consequences of MN formation and rupture. To understand these consequences across biologically relevant contexts, what is needed is a method that can identify and isolate all micronucleated cells from a mixed population in a broad range of cell types and conditions.

Automated detection of MN using conventional image analysis methods is challenging due to the diversity of MN sizes, their morphological similarity to frequently co-occurring nuclear features, including nuclear blebs and chromatin bridges, and their proximity to nuclei. Machine learning methods, specifically deep neural nets, are a powerful approach to discriminate cellular organelles and were used to develop programs to identify nuclei, mitochondria, and spindle poles with high fidelity (*Caicedo et al., 2019b*; *Dang et al., 2023*; *Fischer et al., 2020*). In this study, we apply neural network-based pixel classification to rapidly identify micronucleated cells (VCS MN) or pixel

and instance classification to segment MN with high sensitivity across multiple cell types and imaging conditions (MNFinder). We demonstrate the utility of these analyses for understanding the consequences of micronucleation by combining VCS MN with a phenotype-based cell isolation method, called Visual Cell Sorting (VCS), to define the transcriptomic profile by RNAseq of hTERT-RPE1 cells that have none, intact, or ruptured MN after induced chromosome instability. During visual cell sorting, live single cells expressing nuclear-localized Dendra2 are photoconverted on-demand based on the results of the VCS MN classifier. Specific cell classes are then isolated by FACS (*Hasle et al., 2020*). We show that we can recapitulate an established aneuploidy signature using this method and find that neither micronucleation nor rupture is sufficient to induce significant transcriptional changes. We anticipate that visual cell sorting will be critical to address fundamental questions about how MN contribute to cancer development and enable new high-throughput screens for MN stability and survivability regulators. In addition, we expect MNFinder to have a broad utility for experimenters requiring accurate quantification of MN frequency and characteristics in cultured cells, including those in cancer research, nucleus biology, and virology.

## Results

### Development and training of VCS MN to rapidly classify micronucleated cells by fluorescence imaging

To isolate live micronucleated cells from a mixed population by visual cell sorting, we developed an automated pipeline to rapidly identify nuclei associated with one or more MN. For these studies, we focused on hTERT RPE-1 (RPE1) cells, a near-diploid, non-transformed human cell line that has been extensively used to study chromosome missegregation, aneuploidy, and micronucleation (*He et al., 2019*; *Kneissig et al., 2019*; *Mammel et al., 2021*; *Santaguida et al., 2017*; *Zhang et al., 2015*). We co-expressed a fluorescent chromatin marker (H2B-emiRFP703) to identify nuclei and MN with 3xDendra2-NLS (nuclear localization signal) to identify ruptured MN and photoactivate selected cells (*Hasle et al., 2020*; *Hatch et al., 2013*; *Matlashov et al., 2020*; *Figure 1a*). We refer to this cell line as RFP703/Dendra. In the absence of mitotic perturbations, RPE1 cells have a very low frequency of MN and nuclear atypia, including blebs and chromatin bridges (*Figure 1—figure supplement 1a–b*). Addition of an Mps1 kinase inhibitor (Mps1i), which blocks the spindle checkpoint and accelerates the metaphase-anaphase transition, induces a high level of whole chromosome missegregation and micronucleation (*Garribba et al., 2023*). In RFP703/Dendra cells, Mps1i incubation for 24 hr increased MN frequency from 3% to ~50% of cells (*Figure 1—figure supplement 1b*). Mps1i also increased the frequency of misshapen nuclei and chromatin bridges, but to substantially lower levels than MN (*Figure 1—figure supplement 1b*), and only misshapen nuclei were slightly enriched in the MN +population (*Figure 1—figure supplement 1c*). Most MN +nuclei had only one associated MN and these MN were most likely to contain a single chromosome, consistent with previous analyses (*Figure 1—figure supplement 1d–e*; *Garribba et al., 2023*). Thus, the major effect of short-term Mps1 inhibition on nuclear morphology in RPE1 cells was increased micronucleation, specifically micronucleation of single chromosomes.

The major challenges for classical segmentation of MN using DNA markers alone are the (1) similarity in shape and size of MN to nuclear blebs and chromatin bridges, (2) highly variable MN sizes (1 µm diameter to 10 µm diameter), (3) proximity to the nucleus, and (4) similar chromatin labeling intensities and patterns between MN and nuclei. Therefore, we decided to apply machine learning to overcome current challenges in identifying MN from single-channel fluorescence images. For visual cell sorting, we developed a classifier, VCS MN, based on the widely used U-Net architecture (*Ronneberger et al., 2015*) using Torchvision's pretrained ResNet18 model as its encoder (*Figure 1b*). The goal for its design was to rapidly classify MN-associated nuclei in low resolution images of H2B-emiRFP703 fluorescence for subsequent photoconversion and sorting. A second goal was to capture a large proportion of the cells with small MN close to the nucleus, which represent a highly biologically relevant class of MN that is particularly challenging to segment using classical methods (*Mammel et al., 2021*; *Joo et al., 2023*). To generate training data, one hundred images of RFP703/Dendra cells either untreated or incubated for 24 hr in Mps1i were acquired on a widefield microscope at ×20. To facilitate ground truth annotation and match the 96x96 pixel input required for ResNet18, we used the nucleus segmenter Deep Retina (*Caicedo et al., 2019a*) to generate nucleus masks, which were

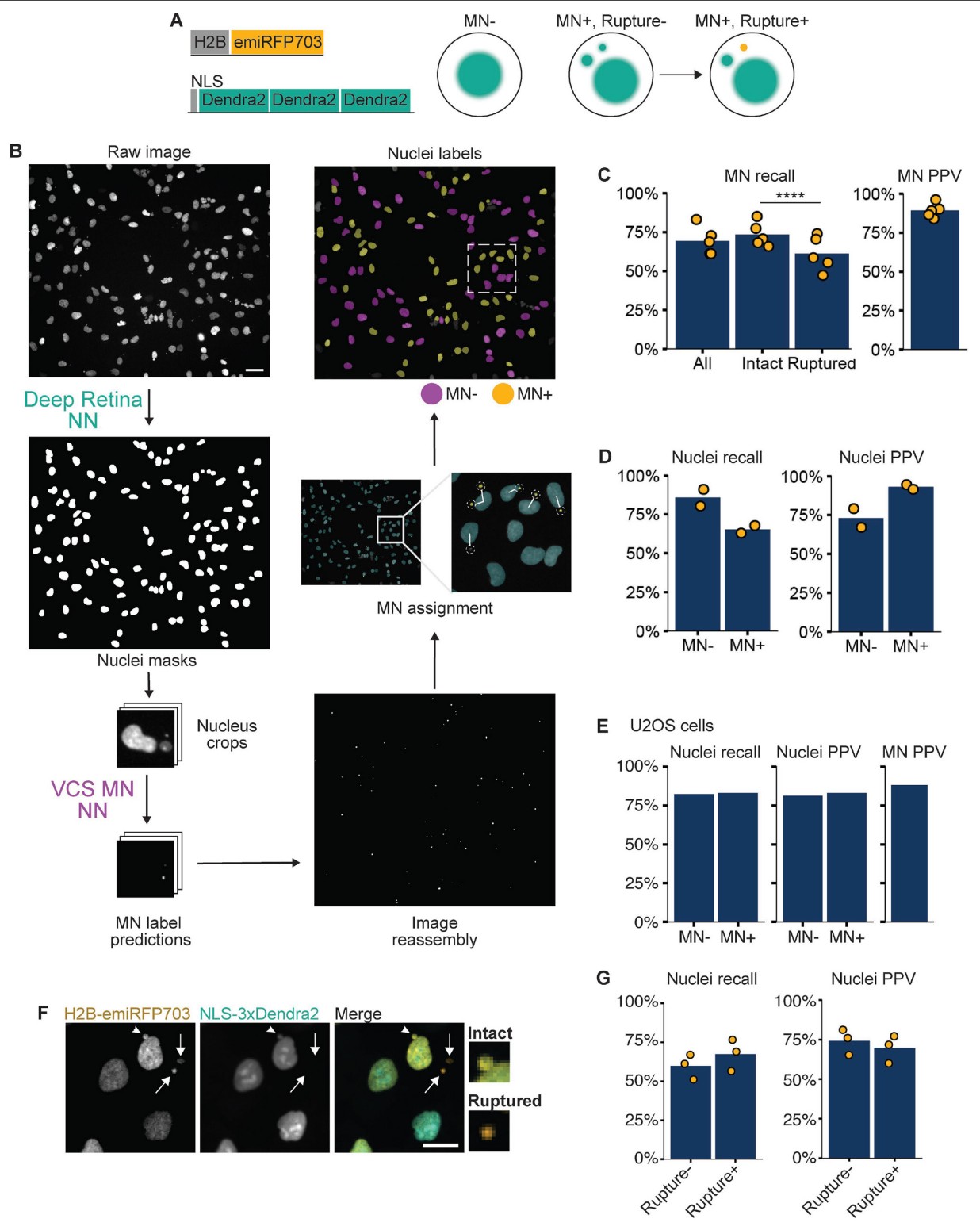

**Figure 1.** VCS MN neural net module identifies micronucleated cells. (**a**) Diagrams of the two constructs transduced into hTERT-RPE1 cells to make the RFP703/Dendra cell line and how the proteins localize in micronucleated (MN+) cells before and after MN rupture. (**b**) Diagram of VCS MN image analysis pipeline on RFP703/Dendra cells incubated in Mps1i. Raw image is of H2B-emiRFP703 taken on a 20 x widefield objective. Nuclei are first segmented using the Deep Retina segmenter and these masks are used to generate image crops centered on each nucleus. Image crops are then resized to fit the ResNet18 UNet model (see *Figure 1—figure supplement 1*) and analyzed by VCS MN to generate MN label predictions, which are mapped back to the raw image. After image reconstruction, MN are assigned to the nearest nucleus based on proximity. Nucleus masks from the Deep

*Figure 1 continued on next page*

*Figure 1 continued*

Retina segmenter are then classified as MN +or MN- to generate the nuclei labels. Scale bar = 40 µm. (**c**) Recall and positive predictive value (PPV) of MN identification within the image crops. MN were manually labeled as intact or ruptured using an image of NLS-3xDendra2 acquired at the same time. N=5, n=264, 158, 365, 283, 249. (**d**) Recall and PPV of MN- and MN +labeled nuclei. N=2, n=328, 186. (**e**) Same as for (d) except images were of U2OS cells expressing H2B-emiRFP703 and 3xDendra2-NLS. N=1, n=85, 95. (**f**) RFP703/Dendra cells with ruptured (arrows) and intact (arrowhead) MN showing a loss of NLS-3xDendra2 signal in ruptured MN. Scale bar = 10 µm. (**g**) Recall and PPV for MN rupture-based nucleus classification. N=3, n=120, 91, 82.

The online version of this article includes the following figure supplement(s) for figure 1:

**Figure supplement 1.** Analysis of nuclear and MN characteristics after Mps1i addition to RPE1 RFP703/Dendra cells.

**Figure supplement 2.** Description of VCS MN classifier training and assessment of MN assignment accuracy.

**Figure supplement 3.** Analysis of VCS MN classification accuracy.

then used to crop each field into smaller images centered around each identified nucleus. MN pixels were then manually annotated on each image crop and labeled as either intact (3xDendra-NLS+) or ruptured (3xDendra-NLS-; *Figure 1—figure supplement 2a*). Analysis of the Deep Retina results found that it frequently missed highly lobulated MN, resulting in an image set with a higher mean nuclear solidity than observed for bulk Mps1i treated cells (*Figure 1—figure supplement 2b*). In total, approximately 2000 single-nucleus crops were used as training data, with 164 crops held back for validation and a further 177 for testing. Because the ResNet18 model takes a three-channel image as input, nuclear crops were expanded to three-channels by duplicating the H2B-emiRFP703 channel and adding a third channel containing the results of Sobel edge detection (*Figure 1—figure supplement 2c*).

The initial output of VCS MN are MN label predictions for each image crop (*Figure 1b*). To identify nuclei associated with at least one MN, these predictions are converted to masks that are mapped back onto the whole field image. MN objects are then assigned to 'parent' nuclear masks by proximity. Nucleus masks associated with at least one MN are labeled as MN +and those associated with no MN are labeled as MN-. MN over 40 pixels (26 µm) away from any nuclear mask (~10% of objects) or within a nucleus mask are discarded as likely false positives signals from cellular debris or inhomogeneity in H2B-emiRFP703 fluorescence.

The accuracy of assigning all MN to the closest nucleus was quantified using the dim cytoplasmic 3xDendra-NLS signal to define the cell boundaries and ground truth MN-nucleus associations (*Figure 1—figure supplement 1a*). Quantification of correct and incorrect nuclei assignments found that the nearest nucleus mask to the MN was in the same cell 97% of the time (*Figure 1—figure supplement 2d–e*). Cells in the training and testing datasets were imaged at subconfluent densities varying between 10% and 50% confluency (*Figure 1—figure supplement 2f*), which is optimal for single cell photoconversion (*Hasle et al., 2020*). In these conditions, MN were on average 3.3 times closer to the nearest nucleus compared to the next nearest one (*Figure 1—figure supplement 2g*). However, 10% of nuclei had borders less than 6 µm apart (*Figure 1—figure supplement 2h*), suggesting that proximity could robustly MN-nucleus pairs even at high cell densities.

## VCS MN classifier performance

VCS MN model performance was determined by analyzing the pixel overlap between segmented MN and the ground truth annotations on the test set. A single pixel overlap between the predicted and actual MN masks was sufficient to call an MN as a true positive since any MN signal classified the adjacent nucleus as MN+. We quantified the positive predictive value (PPV, the proportion of true positives, i.e. specificity) and recall (the proportion of MN found by the classifier, i.e. sensitivity) of VCS MN. MN were identified with a recall of 70% and a PPV of 89% (*Figure 1c*). This translated to a recall of 86% and 65%, respectively, for MN- and MN +nuclei, and a PPV 73% and 93% (*Figure 1d*). Together these indicate that the MN +nuclei class is highly specific for micronucleated cells and suggest that some MN +nuclei are misclassified as MN-, likely due to VCS MN missing the MN in the initial pixel classification.

We next analyzed the source of MN identification errors by VCS MN. Chromatin bridges and nuclear blebs were rarely misidentified as MN (*Figure 1—figure supplement 3a*). Instead, false positives were most often dim and likely out-of-focus objects (*Figure 1—figure supplement 3a–b*) that were misidentified as MN. False negatives were the much larger error class and several features were

enriched in missed MN, including low H2B-emiRFP703 fluorescence intensity, very small or very large size, and being either very close or very far away from the nucleus (*Figure 1—figure supplement 3a–b*). Consistent with smaller MN being missed, we also observed a small, but statistically significant, reduction in recall for ruptured MN (*Figure 1c*), which compact after rupture (*Hatch et al., 2013*; *Vietri et al., 2020*).

To determine whether VCS MN could achieve similar specificity and sensitivity in another cell line, we performed additional training on images of micronucleated cells in other cell types, including U2OS, HeLa, and human fetal fibroblasts (HFF). For testing, U2OS cells expressing H2B-emiRFP703 and 3xDendra2-NLS were treated with Mps1i to induce MN and imaged and annotated as for the RPE1 RFP703/Dendra cells. Analysis of this dataset yielded recall values of 82% and 83% for MN- and MN + cells, respectively, and PPVs of 81% and 83%. MN were identified with a specificity of 88% (*Figure 1e*). In summary, VCS MN can automatically identify micronucleated and non-micronucleated cells with high specificity in low-resolution images from multiple cell lines containing a mix of contaminating objects.

## MN rupture classification

To identify nuclei with intact or ruptured MN, we added a step to the VCS MN module to determine whether any MN assigned to a nucleus were ruptured. Ruptured MN were identified based on 3xDendra2-NLS intensity, which is present in intact MN and very dim in ruptured MN (*Figure 1a and f*). The pipeline measures the maximum Dendra2 green fluorescence intensity in the nucleus and associated MN and MN with a signal less than 0.16 of the nucleus are classified as 'ruptured'. This threshold correctly classified approximately 90% of MN (*Figure 1—figure supplement 3c*). When appended to the VCS MN classifier, this analysis correctly identified 60% of rupture- nuclei (nuclei associated with only intact MN) and 70% of rupture +nuclei (nuclei with at least 1 ruptured MN) with a PPV near 75% in both cases (*Figure 1g*). The lower sensitivity for rupture MN- nuclei is likely because nuclei associated with multiple MN are both more likely to be scored correctly as MN +and have at least 1 ruptured MN (*Figure 1—figure supplement 3d–e*).

## Development of MNFinder for fixed cell MN segmentation

To determine whether the VCS MN classifier could be broadly used to quantify MN frequency, fluorescence, and morphological characterisitics across different imaging modalities and chromatin labels, we analyzed its performance on RPE1 cells treated with Mps1i, fixed and labeled with DAPI, and imaged at ×40 on a confocal microscope. Although nuclear shape and confluency were similar to that in the training set, VCS MN performed substantially worse on the ×40 images (compare to *Figure 1d*) and MN masks failed to cover most of the MN area (*Figure 2a*). Thus, we decided to develop a new classifier for broad MN quantification in fixed cells, called MNFinder.

MNFinder takes a single or multi-channel image of a chromatin-labeled sample at a resolution between 1.5 and 2.8 pixels per µm as input and uses two sets of neural nets to segment nuclei and MN and associate them into 'cells'. MNFinder first crops the image into overlapping 128x128 pixel sections. These are analyzed by (1) an ensemble of three attention-gated U-Nets (Nuc/MN pixel classifier; *Oktay et al., 2018*; *Ronneberger et al., 2015*) and by (2) a multi-pathway UNet3+-based 'cell' instance classifier (*Huang et al., 2020*) based on the architecture described in *Mahbod et al., 2022*. The output masks and labels from both U-Nets then undergo additional filtering and morphological transformations before they are combined into a final multi-channel mask of the nuclei, MN and their associations. A default data output lists individual MN and nuclei objects, thire relationships, and several object features. Additional outputs include multi-channel fluorescence intensity and a wide range nuclear and MN morphometry available via the python scikit tool.

Nucleus and MN pixel segmentation is organized as follows: crops are converted into 2-channel images by adding the result of Sobel filtering to the raw image. These images are then fed into two U-Net inputs, one incorporating multiscaling downsample blocks (*Su et al., 2021*) in the encoder pathway and one without. The MN predictions from both U-Nets are then combined and fed into a third U-Net classifier to produce a consensus output (*Figure 2—figure supplement 1a*). Each of the three neural nets were trained separately and the ensembling neural net was trained with fixed weights set for each input neural net (*Figure 2b*, top). We chose this architecture because incorporating multiscale downsample blocks identified some MN otherwise missed, but produced an overall

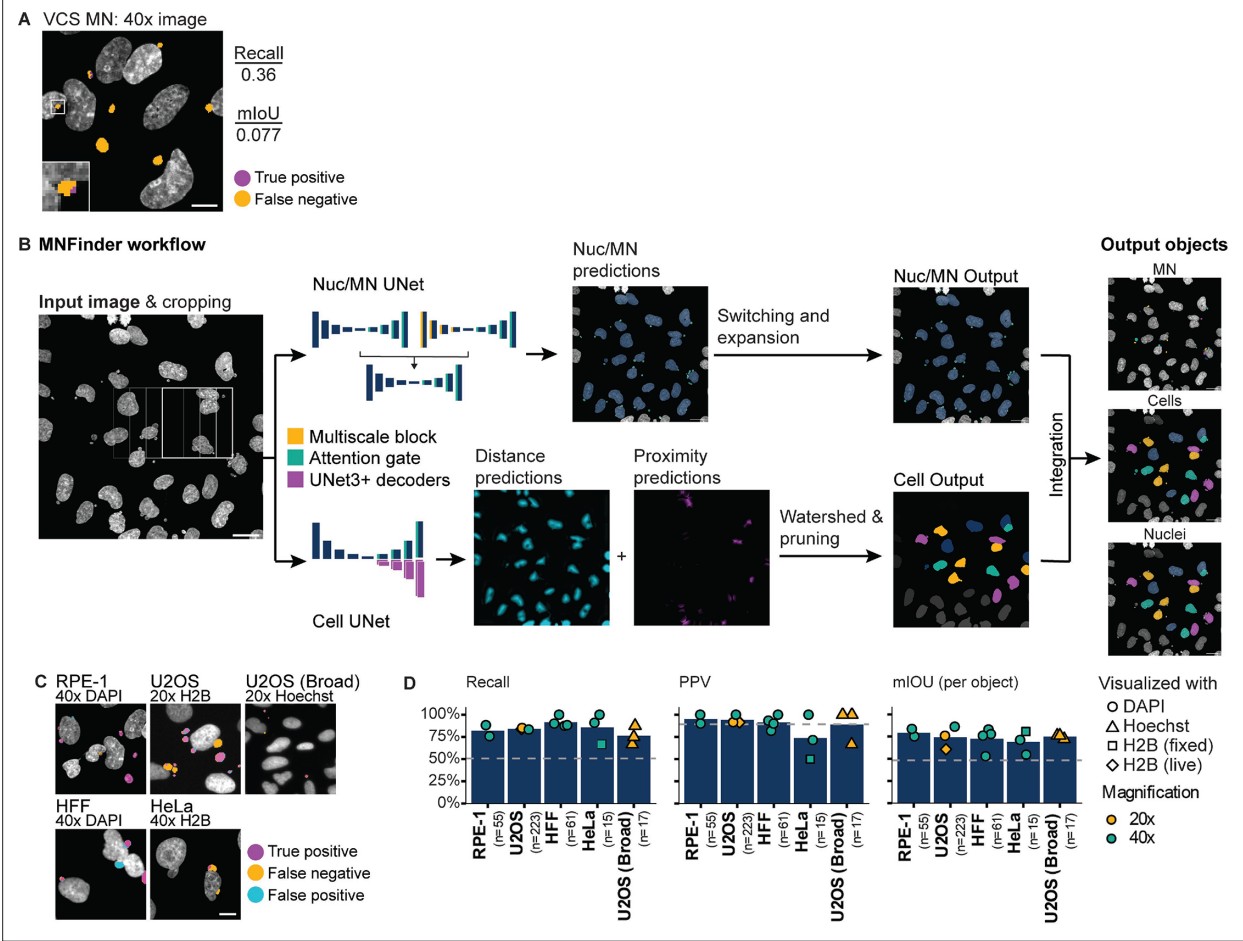

**Figure 2.** MNFinder robustly segments MN across cell types and imaging conditions. (**a**) Representative image showing undersegmentation and low recall by the VCS MN neural net on DAPI-labeled RPE1 cells imaged at ×40 magnification after Mps1i incubation. Images were scaled down prior to VCS MN analysis to match ×20 pixel size. Mean intersection-over-union (mIoU) and recall quantified from N=2, n=33, 31 images. Scale bar = 10 μm. (**b**) Overview of MNFinder module for classifying and segmenting MN and nuclei. Raw images of chromatin are first tiled by a sliding window and then processed by two groups of neural nets: one classifies pixels as either nuclei or MN (Nuc/MN) and one is an instance classifier for cells, with cells being defined as the smallest object that encloses a nucleus and its associated MN. Nuc/MN classifier results are post-processed to reassign MN that were misclassified as small nuclei and to expand MN masks. The cell instance classifier outputs gradient maps, which are used to define cells through watershed-based post-processing (see *Figure 2—figure supplement 1*). Nuc/MN results are integrated with cell results to produce final labels of individual MN, nuclei, and their associations. Image crops are reassembled onto the final image by linear blending. Scale bar = 40 μm. (**c–d**) Example images and MN pixel predictions using MNFinder on multiple cell types (RPE1 H2B-emiRFP703, U2OS, U2OS H2B-emiRFP703, HeLa H2B-GFP, and HFF), chromatin labels (DAPI, H2B-FP), and magnifications (×20, ×40). Scale bar = 10 μm. MN recall, PPV, and mIOU were quantified and averaged per image for each condition. Dotted line=performance of the VCS neural net on RPE1 H2B-emiRFP703 ×20 images. MNFinder performance is similar across conditions except images of H2B-GFP in fixed HeLa cells. N=1. n (on graph)=cells.

The online version of this article includes the following figure supplement(s) for figure 2:

**Figure supplement 1.** Details of UNet architectures and object processing by the MNFinder module.

**Figure supplement 2.** Schematic of training data generation and use in MNFinder classifiers.

**Figure supplement 3.** Analysis of MNFinder errors.

reduction in performance. Using an ensembling neural net allowed us to retain these MN while maintaining the high peformance observed with a single U-Net (*Table 1*). To adjust for the imbalanced nature of our data, where MN make up only a small proportion of any given image, we incorporated attention gates in the upsampling blocks and used the Focal Loss function to direct training to the complex parts of the image (*Lin et al., 2018*).

The 'cell' instance classifier is organized as follows: crops are fed into a U-Net with two U-Net3 +decoder (*Huang et al., 2020*) pathways organized in parallel: one that predicts distance from

**Table 1.** Comparison of neural net model metrics for MN segmentation across cell lines.

| Model | PPV (%) | | | | | Recall (%) | | | | | mIoU, per object (%) | | | | |
|---|---|---|---|---|---|---|---|---|---|---|---|---|---|---|---|
| Cell type | RPE-1 | U2OS | HeLa | HFF | U2OS (Broad) | RPE-1 | U2OS | HeLa | HFF | U2OS (Broad) | RPE-1 | U2OS | HeLa | HFF | U2OS (Broad) |
| UNet | 89.3 (±4.5) | 88.2 | N/A | N/A | 44.8 | 50.8 (±6) | 52 | N/A | N/A | 59.1 | 48.1 (±1.7) | 40.8 | N/A | N/A | 33.7 |
| Attention | **100.0** (±0) | 92.6 (±6.5) | 63.9 (±12.7) | **90.8** (±7.3) | **88.9** (±19.2) | 85.8 (±7.3) | 82.8 (±2.5) | **85.6** (±17.1) | **92.7** (±5.4) | 76.4 (±10.5) | 78.7 (±6.5) | **75.0** (±13.6) | 69.0 (±13.7) | **74.4** (±13.3) | 75.5 (±2.8) |
| MSAttention | 94.8 (±7.3) | 92.2 (±6.8) | 68.8 (±27.2) | 87.6 (±6.6) | 87.9 (±21.0) | 84.4 (±0.7) | **83.2** (±2.75) | 74.4 (±13.5) | 92.3 (±6.4) | **80.6** (±12.0) | 75.2 (±0.4) | 73.6 (±9.2) | **74.9** (±8.4) | 71.3 (±13.4) | **75.6** (±4.0) |
| Ensemble | **100.0** (±0) | **93.8** (±5.4) | **75.0** (±25.0) | 89.9 (±8.74) | **88.9** (±19.2) | 82.6 (±7.4) | 82.8 (±2.5) | **85.6** (±17.1) | **92.7** (±5.4) | 76.4 (±10.5) | **79.3** (±6.0) | 74.2 (±12.8) | 68.8 (±13.1) | 72.4 (±13.4) | 74.8 (±2.3) |

the cell center, where a 'cell' is defined as the smallest convex shape encompassing a nucleus and its associated MN, and one that predicts proximity to neighboring nuclei. A third decoder pathway predicts foreground pixels and is used as input for the first two decoder pathways, an architecture that we found substantially improved training speed and model performance (*Figure 2b*, bottom). To reduce neural net complexity, concatenation operations were replaced with addition (*Lu et al., 2022*) and the foreground/ background decoder pathway foregoes the additional skip connections found in U-Net3 +designs (*Figure 2—figure supplement 1b*).

The outputs from the nucleus/MN and cell classifiers are further processed to improve segmentation and nucleus/MN association accuracy. Masks from the nucleus/MN segmenter are passed to a filter that reclassifies nuclei with an area below 250 px as MN and then expands MN mask boundaries to their calculated convex hull to improve pixel capture (*Figure 2—figure supplement 1c*). 'Cell' masks are derived from the cell instance classifier by first summing the distance and proximity predictions and using the output as seeds for watershed segmentation. To correct for oversegmentation, boundary pixels in these objects are then filtered to retain those that touch either a background pixel or a pixel in the skeletonized proximity map (*Figure 2—figure supplement 1d*). After postprocessing, the results of the nucleus/MN pixel classifier and the 'cell' instance classifier are integrated to produce a final set of labels and masks and a data file (*Figure 2b*).

To train, validate, and test MNFinder, we generated a dataset of 46 images of multiple cell lines, including RPE1, U2OS, HeLa, and human fetal fibroblasts (HFF), after incubation in Msp1i, labeled with dyes or a fluorophore-tagged histone, and imaged at multiple magnifications on multiple microscopes as either a single section or a maximum intensity projection (*Table 2*), after incubation in Mps1i for 24 hr, imaged on multiple microscopes at multiple resolutions. Nuclei and MN pixels were manually annotated on each image to generate the ground truth dataset. To train the nucleus/MN pixel classifier, thirty images with their annotations were cropped, augmented by translation, rescaling, rotation, and flipping, and then converted to 2-channel images to generate a final set of 8800 images. Training was performed on 65% of the image collection, with 7% held back for validation, and the final 28% used for testing (*Figure 2—figure supplement 2a*).

To train the 'cell' instance classifier, annotations of the foreground pixels, distance map, and proximity maps of the full field image were generated on the same starting images by drawing the smallest convex shape that enclosed a nucleus and any associated MN, based on cytoplasmic 3xDendra2-NLS signal or another cell marker. These convex hulls were then converted to distance and proximity maps by scaling the pixel intensity based on distance from the hull's center and distance to adjacent hulls, respectively. These annotated images were then cropped, augmented, and split into training, validation, and test data sets as for the pixel classifier (*Figure 2—figure supplement 2b*).

MNFinder model performance was determined using the test dataset and publicly available images of unperturbed U2OS cells from Broad Bioimage dataset BBBC039v1. We calculated that the unperturbed U2OS cells had an MN frequency of ~8%, compared to the frequency of ~30–70% in the rest of our dataset (*Table 3*). MNFinder showed significant improvement in recall over VCS MN for all image types with an additional improvement in PPV for 3/5 conditions (*Figure 2c–d*). Importantly, recall and PPV were largely insensitive to image resolution, DNA label, and cell type. HeLa H2B-GFP ×40 images were an outlier in terms of performance for unclear reasons, but also had the lowest MN frequeny. We calculated the per object mIoU to determine the quality of the segmentation and found that most MN were accurately segmented with mIoU values between 74% and 79% despite the median MN area being only 34 pixels after resizing (*Figure 2d*, *Table 1*).

Analysis of MNFinder model errors found that chromatin bridges and nucleus blebs were rarely misclassified as MN (*Figure 2—figure supplement 3a*). Instead, highly lobulated nuclei were the main source of false positive errors (*Figure 2—figure supplement 3a, c*). Although small MN were enriched in the false negative error class (*Figure 2—figure supplement 3a–b*), over 75% of MN under 5 $\mu m^2$ were properly identified. These metrics indicate that MNFinder provides accurate and robust MN segmentation across multiple cell lines and image acquisition settings. MNFinder identifies MN with similar sensitivity and substantially improved specificity compared to existing MN enumeration programs (*Table 4*; *Ibarra-Arellano et al., 2024*; *Pons and Mauvezin, 2024*) and is the only one to report a high mIoU, which is necessary for quantifying MN characteristics. This module is available as a Python package, mnfinder, via pip and on the Hatch Lab GitHub repository, along with the training, testing, and validation images and ground truth labels.

**Table 2.** List of image sets used for training, validating, and testing MNFinder.

| Cell_type | DNA_Label | Integrity_label | Max_Int_Proj | Objective mag. | Microscope | Initial image resolution (px/um) | Scaled_image_resolution | Number_images | Training | Validation | Test |
|---|---|---|---|---|---|---|---|---|---|---|---|
| HeLa | H2B-GFP | None | Y | 40 | Confocal_LSM | 5.6 | 2.8 | 1 | | | 1 |
| HeLa | DAPI | None | Y | 40 | Confocal_LSM | 5.6 | 2.8 | 6 | 3 | 1 | 2 |
| HeLa | DAPI | None | N | 20 | Confocal_LSM | 2.8 | 2.8 | 4 | 4 | | |
| hTERT-HFF | DAPI | H3K27-acetyl | Y | 40 | Confocal_Spinning_Disk | 3.9 | 1.95 | 4 | | | 4 |
| hTERT-RPE1 | DAPI | 2xRFP-NLS | Y | 40 | Confocal_LSM | 5.6 | 2.8 | 2 | | | 2 |
| hTERT-RPE1 | H2B-emiRFP703 | NLS-3xDendra2 | N | 20 | Widefield | 1.55 | 1.55 | 16 | 14 | 1 | 1 |
| MCC13 | DAPI | None | N | 20 | Confocal_LSM | 2.8 | 2.8 | 3 | 3 | | |
| U2OS | DAPI | H3K27-acetyl | Y | 20 | Confocal_LSM | 2.8 | 2.8 | 1 | | | 1 |
| U2OS | DAPI | H3K27-acetyl | Y | 40 | Confocal_LSM | 5.6 | 2.8 | 1 | | | 1 |
| U2OS | H2B-emiRFP703 | NLS-3xDendra2 | N | 20 | Widefield | 1.55 | 1.55 | 8 | 6 | 1 | 1 |
| | | | | | | | Total: | 46 | 30 | 3 | 13 |

**Table 3.** Quantification of MN frequency (propotion of cells with at least 1 MN) in images used to evaluate MNFinder.

| RPE-1 | U2OS | HeLa | HFF | U2OS (Broad) |
|-------|------|------|-----|--------------|
| 0.706 | 0.623 | 0.286 | 0.5 | 0.078 |

## Visual cell sorting with VCS MN can isolate live micronucleated and non-micronucleated cells

To test whether visual cell sorting could be used to understand how micronucleation contributes to aneuploid cell phenotypes, we first asked whether using visual cell sorting with the VCS MN classifier could yield cell populations enriched in either micronucleated or non-micronucleated cells. Visual cell sorting takes advantage of recent advances in digital micromirror technology coupled with expression of a convertible fluorophore to label multiple populations of cells with single cell accuracy. By illuminating different classes of cells with different lengths of UV (405 nm) pulses, it can generate up to four distinct proportions of converted Dendra2, which can be sorted by FACS (*Hasle et al., 2020*; *Figure 3a*). To optimize labeling and sorting conditions, we mixed RPE1 cells labeled with CellTrace far-red at a 1:1 ratio with unlabeled cells and targeted the two populations with different UV pulses based on thresholding of the far-red fluorescence intensity (*Figure 3—figure supplement 1a*). We found that activating cells with either an 800ms or a 200ms UV pulse allowed us to define two populations with high precision by FACs and with minimal activation of adjacent cells of the wrong class (*Figure 3—figure supplement 1b*). Reanalysis of sorted populations demonstrated that our protocol yielded a 95% sorting efficiency (*Figure 3—figure supplement 1c*).

We next tested whether Dendra2 red:green ratios were stable for the duration of a cell isolation experiment, based on the estimated time required for imaging, analyzing, and converting multiple six-well plate wells. RPE1 RFP703/Dendra cells were randomly converted with either a 200ms or 800ms UV pulse and nuclear green and red fluorescence intensities quantified at the same positions 0, 4, and 8 hr after conversion. Analysis of the red:green ratios found that, although the intensity of the red fluorescence decreased over time (*Figure 3—figure supplement 1d*), the two activated populations and a small unactivated one (arising from unconverted cells moving into the field of view) remained distinct through the 8 hr time point (*Figure 3b*). These data demonstrate that the change in Dendra red:green ratio after photoconversion persists long enough for accurate cell isolation at the end of a multi-well experiment.

We next asked whether VCS MN classification plus visual cell sorting could accurately activate and isolate MN +and MN- cells. RPE1 RFP703/Dendra cells were incubated with Mps1i for 24 hr prior to imaging to generate MN and Cdk1i was added 1 hr prior to imaging to prevent mitosis, which dilutes the Dendra2(red) signal and frequently leads to MN + cells becoming MN- daughter cells (*Hatch et al., 2013*; *Zhang et al., 2015*). Nuclei were classified by VCS MN as either MN +or MN-, converted with 200ms and 800ms pulses, respectively, and isolated by FACS. Sorted cells were replated in medium containing Cdk1i, fixed, and MN frequency quantified manually. Comparison of classifier PPV for MN +and MN- cells during activation and MN +and MN- cell frequency after FACs found a strong enrichment for the correct cell type in each group, with the increased purity of MN +classified cells being retained during sorting (*Figure 3c*). An increase in MN- cells in the MN +group after sorting is likely due to limitations in sorting efficiency (*Figure 3—figure supplement 1c*) and cells overcoming the Cdk1i block during sorting, plating, and entering mitosis and either dying or producing MN- daughter cells.

**Table 4.** Comparison of MNFinder to existing programs that quantify MN frequency in adherent cell images.

| Study | Cell lines tested | Multiple imaging conditions assessed? | PPV | Recall | Only MN identified? | mIoU |
|-------|-------------------|----------------------------------------|-----|--------|---------------------|------|
| *Pons and Mauvezin, 2024*; QATS | U2OS and HeLa cells | No | 99% | 60% | Yes | ND |
| *Ibarra-Arellano et al., 2024*; micronuclAI | Multiple human and mouse cell lines | Yes | 93.20% | 98.70% | No, nuclear buds classified as MN | ND |
| This study; MNFinder | Multiple human cell lines | Yes | 75.0–100% | 83.2–92.7% | Yes | 74.4–79.3% |

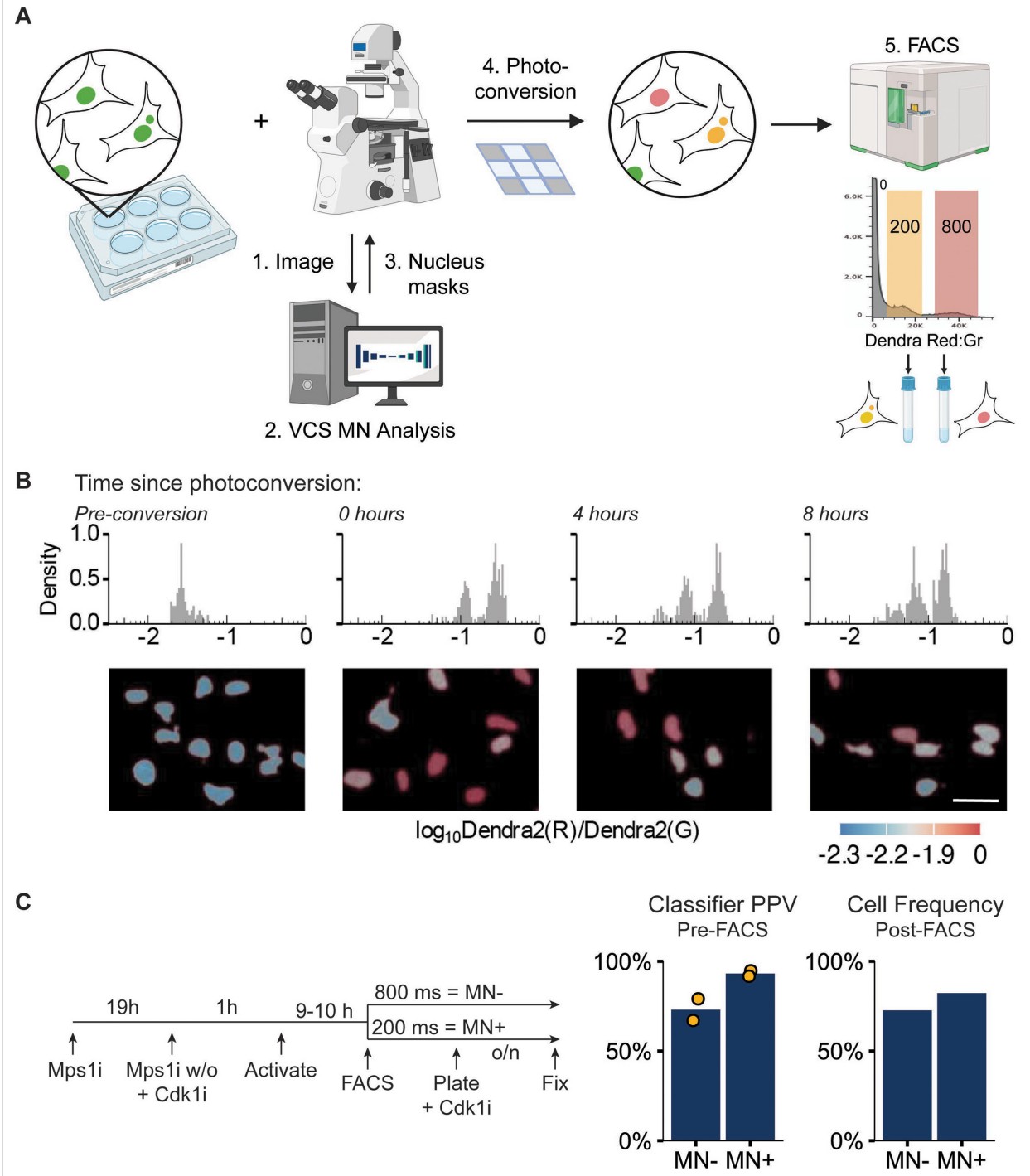

**Figure 3.** Visual cell sorting can isolate RPE1 RFP703/Dendra micronucleated cells. (**a**) Overview of visual cell sorting protocol. Cells are plated in multi-well plates, cellular phenotypes are quantified on-demand during imaging by VCS MN, and labeled nuclei (e.g. MN +and MN-) are photoconverted for either 200 or 800ms, yielding two different ratios of red:green fluorescence (red and yellow nuclei). These differences are quantified by FACs and gated on the red:green ratios for cell sorting. Graphic created with BioRender.com. (**b**) Histogram of nuclear red:green ratio quantification measured on repeated imaging of the same fields 0, 4, and 8 hr after photoconversion. Representative images are pseudo-colored by $\log_{10}$ Dendra red:green ratio (below). N=1, n=82, 353, 285, 313. Scale bar = 40 µm. (**c**) Design of MN cell isolation validation experiment. VCS MN classifier PPV calculated on images acquired during activation and frequency of MN- or MN + cells manually quantified in cells plated and fixed post-FACs sorting. Pre-FACS: N=2, n=251, 263. post-FACS N=1, n=338, 353.

The online version of this article includes the following figure supplement(s) for figure 3:

**Figure supplement 1.** Controls for VCS MN isolation experiments.

To determine how MN frequency affects our ability to accurately isolate MN + cells, we estimated sorted population purity of untreated U2OS cells, which we quantified as having 7.8% MN + cells (*Table 3*). We combined our previously obtained PPV and recall values with a predicted 95% sorting accuracy. The decreased frequency of MN +nuclei meant that the proportion of false positives appeared larger, which decreased purity of the MN +population compared to Msp1i treated RPE1 cells (*Figure 3—figure supplement 1e*). However, the MN +population had a nearly 7-fold enrichment of MN + cells compared to the starting population, similar to the enrichment of tumor cells in patient biopsies (*Wu et al., 2021*) that have been the basis of cancer driver identification by transcriptomic and genetic analysis. Thus, our data strongly suggest that visual cell sorting can be combined with the VCS MN classifier to generate cell populations enriched for MN- and MN + cells from a variety of conditions and MN frequencies that are highly suitable for multiple downstream applications, including genetic screening, bulk RNA and proteomic analyses, and single cell sequencing.

## Validation of visual cell sorting for transcriptome analysis of aneuploid cells

To validate visual cell sorting for aneuploidy-associated transcriptome analysis, we used RNASeq to define gene expression changes in RPE1 cells after Mps1i incubation and optical isolation. Cells were incubated in either DMSO or Mps1i for 24 hr and each population was randomly activated with short and long UV pulses using nuclear masks generated by the Deep Retina segmenter (*Figure 4a*). Conversion of 1500–2000 fields at ×20 magnification allowed us to isolate 13 k cells after FACS for each photoconverted population in each condition. Isolated populations were then processed for RNASeq. Principal component analysis (PCA) of the results revealed that, as expected, results clustered first by treatment group (*Figure 4b*). Analysis of the DMSO samples found minimal differences in gene expression associated with UV pulse duration (*Figure 4—figure supplement 1a–b*), consistent with previous results (*Hasle et al., 2020*). Therefore, data from cells activated at both pulse lengths were pooled in subsequent analyses.

Analysis of gene expression differences identified 2200 differentially expressed genes (DEGs) in Mps1i versus DMSO treated cells, 63 of which had absolute changes >1.5-fold (*Figure 4c*, *Figure 4—figure supplement 1b-c*). We used gene set enrichment analysis (GSEA) to compare these genes to those previously identified in RPE1 cells after induction of chromosome missegregation (*Figure 1—figure supplement 1d-e*; *He et al., 2018*; *Santaguida et al., 2017*). This analysis found substantial overlap between the Hallmark pathways changing in Mps1i cells isolated by visual cell sorting and in previous studies (*Figure 4—figure supplement 1f-h*), including increased expression of inflammation, EMT, and p53 associated genes (*Figure 4d*). Additional DEGs observed in visual cell sorting samples fell into similar function categories and potentially resulted from increased sequencing depth and increased sensitivity from the exclusion of highly lobulated nuclei from our analysis by the Deep Retina segmenter. These data confirm that visual cell sorting can accurately and sensitively identify biologically relevant transcriptional changes in aneuploid RPE1 cells.

## MN rupture induces few transcriptional changes

MN formation frequently coincides with increased aneuploidy, as both a cause and a consequence. Therefore, we asked whether the presence or rupture of an MN contributed to the genome-wide aneuploidy transcriptional response (*Andrade et al., 2023*; *He et al., 2018*; *Santaguida et al., 2017*). Although it has been suggested that MN rupture induces rapid upregulation of inflammatory genes via activation of cGAS-STING signaling, initial and subsequent studies have demonstrated that this response does not occur during the first cell cycle and also can occur in RPE1 cells that lack cGAS via a RIG-I dependent mechanism (*Chen et al., 2020*; *Flynn et al., 2021*; *Harding et al., 2017*; *Mackenzie et al., 2017*; *Mohr et al., 2021*; *Sato and Hayashi, 2024*). Thus, we do not expect the lack of cGAS expression in RPE1 cells to alter MN transcriptional responses.

We first analyzed gene expression changes in MN +versus MN- RPE1 cells after Mps1i treatment (*Figure 5a*). Quantification of nuclear atypia in the masked nuclei found no difference in the proportion of MN +nuclei with lobulated nuclei, chromatin bridges, or rounded up nuclei compared to MN- ones (*Figure 5—figure supplement 1a*), likely due to exclusion of most of these nuclei by the Deep Retina segmenter (*Figure 1—figure supplement 1b*). RNASeq results clustered largely by replicate rather than by the presence of an MN (*Figure 5b*). Consistent with a lack of a strong transcriptional effect,

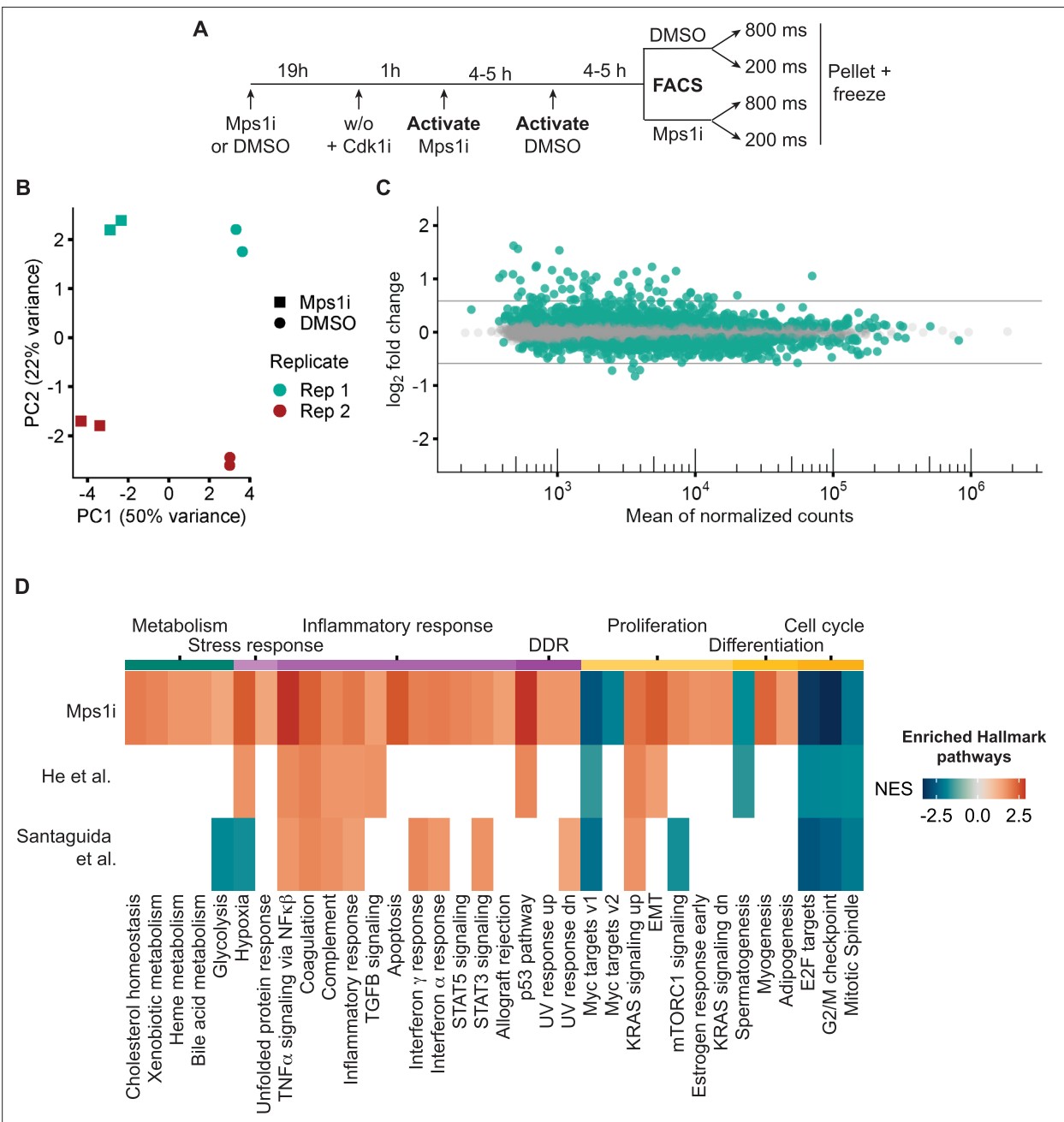

**Figure 4.** Visual cell sorting pipeline identifies Mps1i transcriptional response. (**a**) Outline of experiment. Nuclei identified by the Deep Retina segmenter were randomly activated for 800 or 200ms. (**b**) PCA plot showing clustering of Mps1i-treated and DMSO-treated cells by treatment (major) and by replicate (minor). Each experimental replicate represents two technical replicates. (**c**) MA plot of genes identified by RNASeq. Differentially expressed genes (FDR adjusted p-value <0.05) are in green. Gray lines represent 1.5-fold-change in expression. (**d**) Heatmap of Hallmark pathway enrichment between visual cell sorting data and data from *Santaguida et al., 2017* and *He et al., 2019* analyses of RPE1 cells after chromosome missegregation. Hallmark pathways (bottom) were clustered based on manually annotated categories (top).

The online version of this article includes the following figure supplement(s) for figure 4:

**Figure supplement 1.** Differential UV pulses do not induce substantial transcriptional changes.

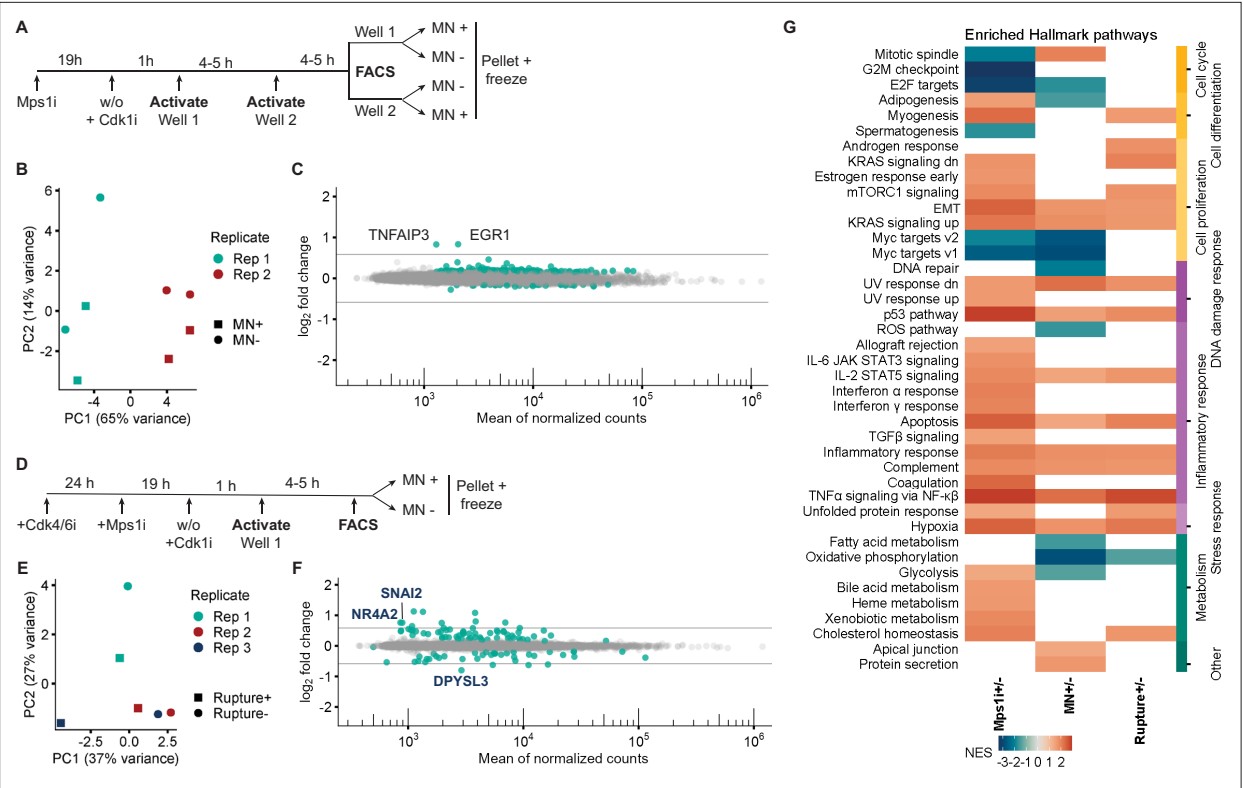

**Figure 5.** Micronucleation and rupture result in few transcriptional changes. (**a**) Outline of experiment for MN +and MN- cell isolation from Mps1i treated RFP703/Dendra cells. (**b**) PCA plot showing clustering of MN + and MN- cells by replicate (major) and condition (minor). (**c**) MA plot of genes identified in MN +/-RNASeq. Of the identified DEGs, only two have fold-changes larger than 1.5. Both, TNFAIP3 and EGR1, are also significantly upregulated in Mps1i-treated cells. (**d**) Outline of experiment for MN rupture + and rupture- cell isolation. (**e**) PCA plot showing clustering of intact MN and ruptured MN cells by condition and replicate. (**f**) MA plot of genes identified in MN rupture +/-RNASeq. The three DEGs with a fold-change higher than 1.5 and unique to this dataset are labeled on the plot. (**g**) Heatmap of Hallmark pathway enrichment in datasets of DMSO vs Mps1i, Mps1i-treated cells with and without MN, and synchronized, Mps1i-treated, MN + cells with and without MN rupture. All DEGs, defined as an FDR less than 0.05, are included. Pathways are grouped based on manual annotation (right). 6/43 pathways are unique to the MN +/- profile compared to Mps1i+/-, and 2/43 pathways are unique to MN rupture +/-compared to Mps1i+/-.

The online version of this article includes the following figure supplement(s) for figure 5:

**Figure supplement 1.** Analysis of visual cell sorting MN images.

DEG analysis identified few statistically significant changes in gene expression (*Figure 5c*, *Figure 5—figure supplement 1i*). The only two genes that were upregulated more than 1.5-fold, EGR1 and TNFAIP3, were also strongly upregulated in the Mps1i bulk analysis (*Figure 5—figure supplement 1cj*). Together, these results suggest that micronucleation does not drive substantial changes in the transcriptional program of aneuploid cells.

We next asked whether MN rupture affected transcription by using visual cell sorting to isolate MN + cells with all intact MN (rupture- nuclei) and MN + cells with at least 1 ruptured MN (rupture +nuclei). To minimize the number of MN rupturing during the experiment, and thereby decreasing the purity of the intact MN population, we synchronized the cells prior to Mps1i treatment and converted them in the following G1 phase (*Figure 5—figure supplement 1b*). Analysis of MN rupture by live-cell imaging found this led to only about 10% of intact MN cells becoming ruptured MN cells over 4 hr, the time to photo-convert one well of a six-well plate. We estimated this would reduce the intact MN population PPV from 70.6% to 61.9% (*Figure 5—figure supplement 1c*).

Analysis of gene expression changes between intact and ruptured MN + cells by RNAseq showed that samples clustered first by condition (*Figure 5e*) and identified a large number of DEGs, 14 of which had absolute changes greater than 1.5-fold (*Figure 5f*, *Figure 5—figure supplement 1k*). Of these 14, 3 were not previously identified in the Mps1i+analysis (*Figure 5—figure supplement 1l*) and GSEA analysis confirmed that almost all of the pathways altered in either MN +and MN rupture + cells

overlapped with those identified in the bulk Mps1i population (*Figure 5g*, *Figure 5—figure supplement 1m-n*). These data suggest that MN rupture has minimal unique effects on transcription in the first interphase after whole chromosome missegregation.

## MN formation and rupture do not contribute to the aneuploidy transcription signature

Due to the large overlap between DEGs in MN rupture cells compared to the bulk aneuploid population, we tested the hypothesis that MN rupture is the cause of these transcriptional changes. If this is the case, then we expect to observe the changes in gene expression to get stronger as cells with ruptured MN are more enriched in the population. We used k-means clustering to visualize changes in DEG expression between Mps1i treated cells, MN + cells, and MN rupture cells. To minimize noise from potential batch effects in the MN +dataset, we included individual replicates from that and the MN rupture experiment in our analysis. This analysis identified one gene cluster that showed potentially elevated expression in MN rupture cells, which included all the most highly upregulated genes after MN rupture (*Figure 6a*, *Figure 5—figure supplement 1o*).

We next attempted to validate increased expression of two of these genes, EGR1 and ATF3, after MN rupture by immunofluorescence. As a positive control, increased protein levels were observed after addition of hEGF or doxorubicin, a DNA damaging agent, to RPE1 cells (*Figure 6—figure supplement 1a–b*). Both EGR1 and ATF3 showed increased expression after Mps1i treatment, however, no increase in EGR1 and only a small shift in ATF3 was observed in cells with ruptured MN (*Figure 6b–c*). Based on these results, we conclude that MN rupture is strongly unlikely to be a significant driver of transcriptional changes in aneuploid cells.

## Cells with ruptured MN have higher levels of aneuploidy

One explanation for our observation that some genes increase in expression strength in cells with ruptured MN is that this population is enriched in aneuploid cells. We therefore analyzed aneuploidy levels in Mps1i-treated RPE1 cells by quantifying the number of chromosomes 1, 11, and 18 by DNA-FISH. Cells were manually classified as MN- or MN +by DAPI labeling, and MN classified as ruptured based on loss of H3K27Ac labeling (*Mammel et al., 2021*; *Mohr et al., 2021*; *Figure 6d*). Aneuploidy frequency varied between chromosomes but was consistently higher for MN +compared to MN- cells and higher for 2/3 chromosomes for ruptured compared to intact MN + cells (*Figure 6e*). Similar results were obtained when transcription loss due to MN rupture was considered (functional aneuploidy; *Figure 6—figure supplement 1c*). Overall, our results strongly suggest that micronucleated cells likely have slightly higher levels of aneuploidy and that MN formation and rupture do not cause specific transcriptional changes in the first cell cycle.

## Discussion

In this study, we present two machine-learning based modules to identify MN and micronucleated cells from single channel fluorescence images of chromatin or DNA and combine one with visual cell sorting to profile the transcriptional responses to MN formation and rupture. We demonstrate that our MN segmentation pipeline MNFinder can robustly classify and segment MN across multiple adherent cell types and fluorescent imaging conditions. Further, we demonstrate that a separate MN cell classifier, VCS MN, rapidly and robustly identifies micronucleated cells from low resolution images and can be combined with single-cell photoconversion to accurately isolate live cells with none, intact, or ruptured MN from a mixed population. Using this platform, we find that, unexpectedly, neither micronucleation nor rupture triggers gene expression changes beyond those associated with a slight increase in aneuploidy. Overall, our study brings a powerful high-throughput optical isolation strategy to MN biology that enables new types of investigations and a new module for quantifying MN characteristics in adherent cells that will allow naïve users to rapidly and robustly quantify these compartments.

Visual cell sorting isolation of micronucleated cells has several advantages over current methods to identify the mechanisms and consequences of MN formation and rupture. First, it can be used on any adherent cell line. This overcomes challenges involved with using lamin B2 overexpression to inhibit MN rupture, which is limited to specific cell lines and MN types (*Hatch et al., 2013*; *Mammel et al.,*

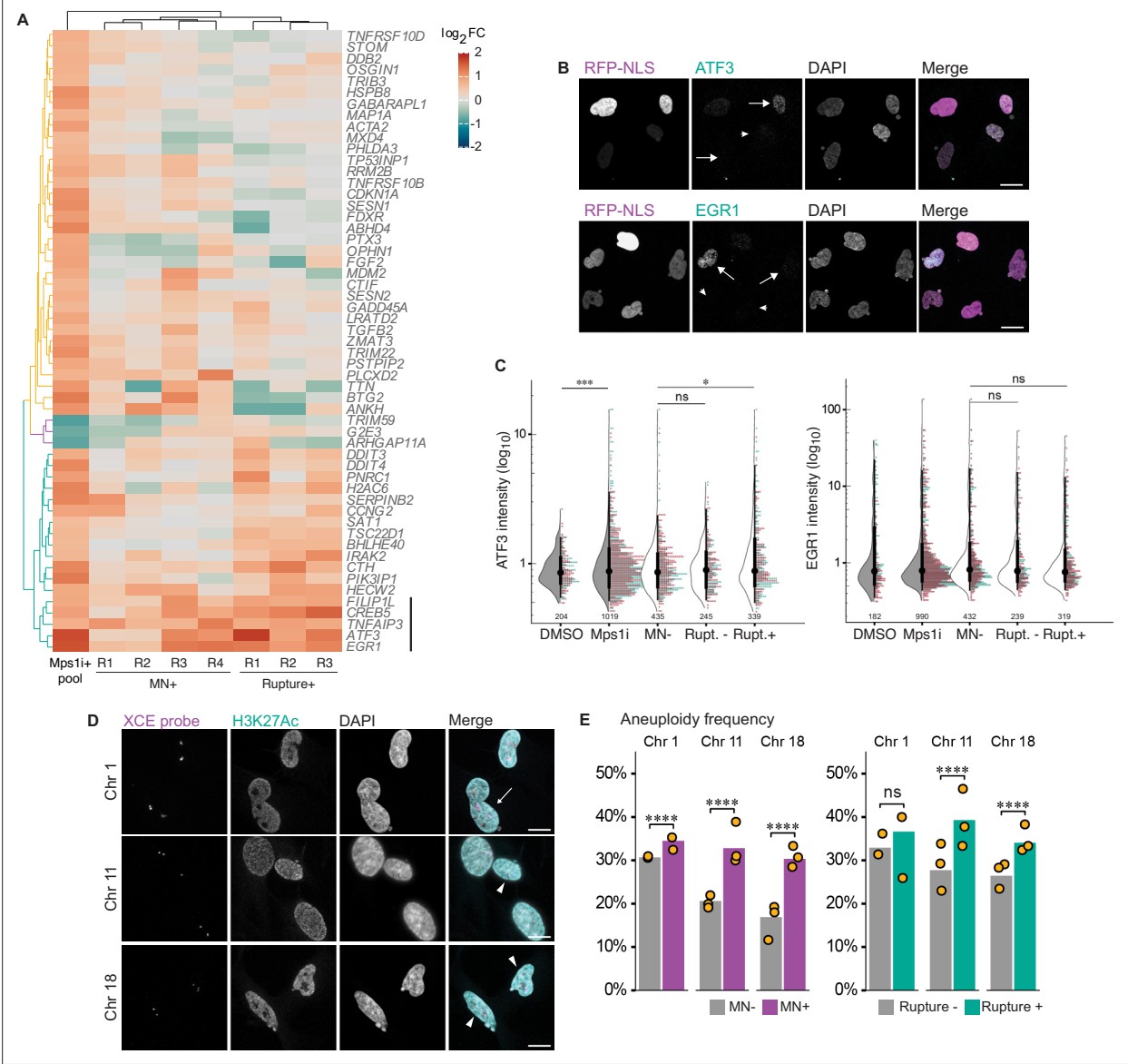

**Figure 6.** Micronucleation and rupture do not significantly alter the aneuploidy transcription response. (**a**) Heatmap of Mps1i+DEGs (cutoff: absolute fold-change ≥1.5) compared to MN +and rupture +replicates. Euclidean distances calculated for features and samples and clustered by complete-linkage. Genes lacking values for one more experiment types were excluded. Line=gene cluster upregulated in MN rupture + cells. (**b**) Representative images and quantification of ATF3 and EGR1 labeling in RPE1 2xRFP-NLS cells after Mps1i incubation. Arrows = ruptured MN, arrowheads=intact MN. Scale bar=20 μm. (**c**) Quantification of normalized ATF3 and EGR1 mean nuclear intensity in manually classified cells. N=2 (graph colors), n=on graph, p: ns >0.05, *≤0.05, ***≤0.001 by GEE. (**d**) Representative images of DNA FISH for chromosomes 1, 11, and 18 and H3K27Ac identification of intact MN. Arrows=ruptured MN, arrowheads=intact MN. (**e**) Quantification of aneuploidy frequency (foci do not equal 2) per chromosome. Cells manually classified as MN- or MN+, and MN rupture- or rupture+. MN: Chr 1: N=2, n=429, 158; Chr 11: N=3, n=406, 313, 160; Chr 18: N=3, n=425, 202, 230. Rupture: Chr 1: N=2, n=187, 74; Chr 11: N=3, n=190, 108, 71; Chr 18: N=3, n=186, 102, 101.

The online version of this article includes the following figure supplement(s) for figure 6:

**Figure supplement 1.** Controls related to *Figure 6*.

*2021*; *Xia et al., 2019*) and is complicated by additional changes in mitosis and gene expression (*Agustinus et al., 2023*; *Han et al., 2020*; *Kuga et al., 2014*; *Pujadas Liwag et al., 2024*). In addition, visual cell sorting overcomes cell line and MN content restrictions present in systems that induce missegregation of single chromosomes or chromosome arms by enabling analysis of all missegregation events in any genetic background (*Lin et al., 2023*; *Ly et al., 2019*; *Ly et al., 2017*; *Shoshani et al., 2021*; *Trivedi et al., 2023*). Unlike live single-cell assays, it is highly scalable and eliminates

selection pressures and restrictions added by clonal expansion (*Mohr et al., 2021*; *Papathanasiou et al., 2023*; *Zhang et al., 2015*). Importantly, VCS MN isolation captures whole live cells, overcoming limitations associated with MN purification (*Agustinus et al., 2023*; *Klaasen et al., 2022*; *Mohr et al., 2021*; *Papathanasiou et al., 2023*; *Tang et al., 2022*), and permits time-resolved analyses of cellular changes and MN chromatin. Visual cell sorting also has an advantage over other optical isolation methods, including image-enabled cell sorting (*Schraivogel et al., 2022*), as it can be adapted to any wide-field microscope by adding a digital micromirror to existing equipment, and can be performed on attached cells, which is critical for accurate MN identification (*Li et al., 2015*).

Visual cell sorting of MN and our MN segmentation modules do have limitations. Due to Dendra2 signal decay and ongoing MN rupture, only about 200,000 live cells can be analyzed and targeted per visual cell sorting experiment with VCS MN. For optical pooled screening or analysis of rare cells, this limits the number of genes or depth of analysis that can be achieved. Cell fixation would overcome this issue and efforts to improve cell sorting and sample extraction after fixation are ongoing (*Kanfer et al., 2021*; *Yan et al., 2021*). Visual cell sorting also requires the introduction of at least one photoconvertible or activatable protein to mark the cells and a second fluorescent protein to identify ruptured MN. This limits the channels available for additional phenotype identification. However, recent advances in cell structure prediction (*Johnson et al., 2023*) may vastly expand the phenotypic information available from limited cell labels. VCS MN and MNFinder precision vary across cell types and widely divergent nuclear morphologies from the training set could impair performance. Additional training of the neural net should improve this metric, as has been the case with other cellular structures (*Caicedo et al., 2019b*; *von Chamier et al., 2021*), but different algorithm architectures will likely be required to identify MN in signal-rich environments like organoids or tissue samples. In these cases, correct association of MN to their parent nuclei may also require using a cytoplasm or cell surface label to define ground truth MN-nucleus associations and/or developing a new module to perform this task.

Comparison of our RNASeq analysis of aneuploid RPE1 cells with previously published datasets demonstrates that visual cell sorting is highly suitable for bulk genomic analyses. We observed a large overlap between upregulated pathways in our study, previous analyses of Mps1i-treated RPE1 cells, and aneuploidy more broadly, including inflammation, endothelial-to-mesenchymal transition, and p53 (*He et al., 2019*; *Santaguida et al., 2017*). We did observe a globally lower change in gene expression levels compared to previous studies. Although the values are within the variation anticipated for different preparation and sequencing protocols, it could also reflect the exclusion of cells with highly lobulated nuclei from our analysis and suggests that nuclear atypia may be a significant contributor to aneuploidy signaling changes. Additional visual cell sorting experiments using nuclear solidity to define cell populations could be used to accurately quantify this contribution.

Our analysis of micronucleated cells and cells with ruptured MN found only a handful of genes that were uniquely upregulated after MN rupture and we were unable to validate a contribution of either condition to protein changes in the cell. One potential interpretation of these results is that MN rupture or micronucleation does cause transcriptional changes, but that they are masked by signals from co-occurring events. Although we cannot rule this out, we favor the alternative interpretation that MN formation and rupture do not lead to biologically relevant transcriptional changes for several reasons. The first is that, of potential co-occurring nuclear phenotypes we quantified, only nuclear lobulation was frequently observed in Mps1i treated cells and was only slightly enriched in micronucleated cells (*Figure 1—figure supplement 1b–c*). Furthermore, this frequency was significantly reduced and MN + cell enrichment was lost when nuclei were classified using VCS MN (*Figure 1—figure supplement 2b*, *Figure 2—figure supplement 2a*). Thus, nuclear atypia, chromatin bridges, and mitotic DNA damage are unlikely masking other transcriptional signals in our RNAseq results. Second, we were unable to validate protein-level increases of two increasingly upregulated genes in cells with ruptured MN by immunofluorescence, suggesting that any missed DEGs are unlikely to be biologically relevant. Third, we were able to identify small changes in aneuploidy gene upregulation in both MN +and ruptured MN cells, strongly suggesting that our analysis was sensitive to small differences in transcript abundance. In addition, we observed an increase in aneuploidy levels in cells with MN and ruptured MN that correlated with the increase in aneuploidy gene expression. Taken together, these data strongly suggest that the most likely explanation is that MN, intact or ruptured, do not induce a meaningful transcriptional signal in the first cell cycle.

A potential limitation of our current data is that the cell line used, RPE1, has reduced cGAS expression. MN rupture has been proposed to activate cGAS/STING signaling (*Bakhoum et al., 2018*; *Harding et al., 2017*; *Mackenzie et al., 2017*) and the strong localization of cGAS to ruptured MN led to a widespread model where this activation occurred in the first cell cycle. However, the original studies did not observe upregulation of interferon-stimulated genes until several days after MN rupture, suggesting that cGAS recruitment to MN does not activate it, and now multiple lines of evidence strongly suggest cGAS bound to ruptured MN is incapable of activating STING (*Flynn et al., 2021*; *Harding et al., 2017*; *Mohr et al., 2021*; *Sato and Hayashi, 2024*). Therefore, we expect that VCS analysis of MN-associated gene expression in a highly active cGAS/STING cell line will yield the same results as what we observed in RPE1 cells.

Visual cell sorting is a highly flexible platform that substantially expands our ability address fundamental questions in MN biology. Visual cell sorting MN isolation can be used for optical pooled screening, an unbiased method that would be ideal to identify mechanisms of MN rupture, and, in combination with dCas9-based chromosome labeling (*Chen et al., 2013*; *Maass et al., 2018*; *Tanenbaum et al., 2014*), to identify mechanisms that enrich specific chromosomes in MN and drive cancer-specific aneuploidies (*Ben-David and Amon, 2020*). Techniques to improve retrieval and sorting of formaldehyde-fixed cells would significantly facilitate these analyses by enabling collection of a larger number of cells per experiment and allowing use of slower but more accurate MN segmentation modules, such as MNFinder, to label cell populations.

A current strength of visual cell sorting with VCS MN is the ability to recover a population of live cells with either none, intact, or ruptured MN. This allows, for the first time, large-scale analyses of post-mitotic genetic and functional changes caused by micronucleation. For instance, aneuploid cells with and without MN could be analyzed for acquisition of disease-associated behaviors, including proliferation and migration, and used in in vivo tumorigenesis and metastasis assays to directly assess their contribution to cancer development. In addition, a similar method developed in the Korbel lab shows the power of this technique used to quantify the frequency and extent of genetic changes after chromosome missegregation (*Cosenza et al., 2024*). The ability of visual cell sorting to use nucleus masks generated by any image analysis program means that the user can employ any open-source MN segmentation module to optimize for higher purity for these experiments, or a broader recall to identify rare populations in single-cell experiments. In summary, automated MN segmentation and visual cell sorting are poised to provide critical insights into a wide range of questions about how MN form, rupture, and contribute to disease pathologies.

# Materials and methods

**Key resources table**

| Reagent type (species) or resource | Designation | Source or reference | Identifiers | Additional information |
|---|---|---|---|---|
| Cell line (*Homo sapiens*) | HEK293T (transformed, kidney) | ATCC | CRL-3216, RRID:CVCL_0063 | |
| Cell line (*Homo sapiens*) | U2OS (transformed, osteosarcoma) | ATCC | HTB-96. RRID:CVCL_0042 | |
| Cell line (*Homo sapiens*) | HeLa (transformed, cervical carcinoma) | ATCC | CRM-CCL-2, RRID:CVCL_0030 | |
| Cell line (*Homo sapiens*) | hTERT RPE-1; RPE1 (immortalized, epithelial) | ATCC | CRL-4000, RRID:CVCL_4388 | |
| Cell line (*Homo sapiens*) | hTERT HFF; HFF (immortalized, fibroblast) | PMID:9817205 | | Cell line maintained in D. Galloway lab. |
| Cell line (*Homo sapiens*) | MCC13 (transformed, Merkel cell carcinoma) | ECACC | 10092302, RRID:CVCL_2583 | Cell line maintained in P. Nghiem lab |
| Cell line (*Homo sapiens*) | HeLa H2B-GFP | Millipore | SCC117, RRID:CVCL_ZM02 | Cell line maintained in D. Avgousti lab |
| Cell line (*Homo sapiens*) | hTERT RPE-1 NLS-3xDendra2, H2B-emiRFP703; RFP703/Dendra | This study | | RPE1 cells used for VCS MN experiments |

*Continued on next page*

*Continued*

| Reagent type (species) or resource | Designation | Source or reference | Identifiers | Additional information |
|---|---|---|---|---|
| Cell line (*Homo sapiens*) | U2OS NLS-3xDendra2, H3B-emiRFP703 | This study | | U2OS cells used for MNFinder training |
| Cell line (*Homo sapiens*) | hTERT RPE-1 NLS-3xDendra2-P2A-H2B-miRFP703 | This study | | RPE1 cells used for VCS MN training |
| Antibody | Anti-gH2AX phospho (Ser139) (mouse monoclonal) | Biolegend | Cat# 613401, RRID:AB_315794 | IF (1:500) |
| Antibody | Anti-ATF3 (rabbit monoclonal) | CST | Cat# 18665, RRID:AB_2827506 | IF (1:400) |
| Antibody | Anti-EGR1 (rabbit monoclonal) | CST | Cat# 4154, RRID:AB_2097035 | IF (1:1600) |
| Antibody | Anti-H3K27Ac (rabbit polyclonal) | Abcam | Cat# ab4729, RRID:AB_2118291 | IF (1:500) |
| Antibody | AF647 goat-anti-mouse (polyclonal, secondary) | Life Technologies | Cat# A21236 | IF (1:1000) |
| Antibody | AF488 goat anti-rabbit (polyclonal, secondary) | Life Technologies | Cat# A11034 | IF (1:2000) |
| Recombinant DNA reagent | pH2B-emiRP703 | Addgene | Cat# 136567, RRID:Addgene_136567 | Verkhusha lab |
| Recombinant DNA reagent | pLVX-EF1a-NLS-3xDendra2-blast (plasmid) | This study | | Lentiviral plasmid based on pLVX-puro backbone (Clontech) to express NLS-3xDendra2 |
| Recombinant DNA reagent | pLVX-EF1a-H2B-emiRFP703-neo (plasmid) | This study | | Lentiviral plasmid based on pLVX-puro backbone (Clontech) to express H2B-emiRFP703 |
| Recombinant DNA reagent | pLenti-CMV-NLS-Dendra2x3-P2A-H2B-miRFP | PMCID:PMC7273721 | | Lentiviral plasmid to co-express H2B-miRFP703 and NLS-Dendra2 |
| Commercial assay or kit | RNAqueous micro | Thermo Fisher | AM1931 | |
| Commercial assay or kit | SMART-Seq v4 ultra-low input RNA | Takara | 634894 | |
| Commercial assay or kit | Nextera XT DNA library preparation | Illumina | FC-131–1024 | |
| Chemical compound, drug | DAPI | Life Technologies | D1306 | (1 µg/mL) |
| Chemical compound, drug | Vectashield | Vector Labs | H-1000 | |
| Chemical compound, drug | RO-3306; Cdk1i | Sigma-Aldrich | SML0569 | (10 µM) |
| Chemical compound, drug | BAY1217389; Mps1i | Fisher Scientific | 501872752 | Msp1 inhibitor (100 nM) |
| Chemical compound, drug | PD-0332991; Cdk4/6i | Sigma-Aldrich | PZ0199 | Cdk4/6 inhibitor (1 µM) |
| Chemical compound, drug | doxorubicin hydrochloride | Fisher Scientific | BP25165 | (2 µg/mL) |
| Chemical compound, drug | hEGF | Peprotech | AF-100–15- | (5 ng/mL) |
| Chemical compound, drug | Phenol-red free DMEM/F12 | GIBCO | 21041025 | Used during VCS imaging |
| Chemical compound, drug | CellTrace Far Red | Thermo Fisher | C34572 | Cell label dye used in VCS validation |
| Software, algorithm | Metamorph (v7.10.1.161) | Molecular Devices | RRID:SCR_002368 | |

*Continued on next page*

*Continued*

| Reagent type (species) or resource | Designation | Source or reference | Identifiers | Additional information |
|---|---|---|---|---|
| Software, algorithm | Leica Application Suite X | Leica | RRID:SCR_013673 | |
| Software, algorithm | VCS MN | This study | | Available at https://github.com/hatch-lab/fast-mn |
| Software, algorithm | MNFinder | This study | | Available at https://github.com/hatch-lab/mnfinder |
| Other | Chromosome 1 XCE – orange (*Homo sapiens*) | MetaSystems | D-0801–050-OR | DNA-FISH probe |
| Other | Chromosome 11 XCE – orange (*Homo sapiens*) | MetaSystems | D-0811–050-OR | DNA-FISH probe |
| Other | Chromosome 18 XCE – orange (*Homo sapiens*) | MetaSystems | D-0818–050-OR | DNA-FISH probe |
| Other | PNA CENPB-Cy5 | PNA Bio | F3005 | DNA-FISH probe |
| Other | Leica DMi8 with Adaptive Focus | Leica | | Microscope for VCS |
| Other | Mosaic 3 Digital Micromirror Device | Andor | | Microscope component for VCS |
| Other | Leica DMi8 laser scanning confocal microscope | Leica | | Confocal microscope |
| Other | CSU spinning disk unit | Yokagawa | | Confocal microscope component |
| other | FACS Aria II | BD Biosciences | RRID:SCR_018934 | Cell sorter |

## Plasmid construction

pLVX-EF1a-NLS-3xDendra2-blast was created by PCR of NLS-3xDendra2 from pLenti-CMV-NLS-Dendra2x3-P2A-H2B-miRFP (*Hasle et al., 2020*) using primer sequences 5'-caagtttgtacaaaaaagtt ggcaccATGG-3' and 5'-**TTA**GGAAAAATTCGTTGCGCCGCTCCC-3', followed by ligation into pLVX-EF1a-blast. pLVX-EF1a-H2B-emiRFP703-neo was created by PCR of H2B-emiRFP703 from pH2B-emiRP703 (a gift from Vladislav Verkhusha, AddGene #136567) with primers 5'-ATGCCAGAGCCA GCGAAG-3' and 5'-TTAGCTCTCAAGCGCGGTGATC-3', followed by ligation into pLVX-EF1a-neo. pLVX-EF1a was derived from pLVX-puro (Clontech) by replacing the CMV promoter with EF1a and replacing the selection marker with blast or neo.

## Cell culture and construction of cell lines

hTERT RPE-1 cells were cultured in DMEM/F12 (Gibco) supplemented with 10% FBS (Sigma), 1% penicillin-streptomycin (Sigma), and 0.01 mg/mL hygromycin B (Sigma) at 5% $CO_2$ and 37 °C. U2OS, hTERT-human fetal fibroblasts (HFF), and HeLa cells were cultured in DMEM (Gibco) supplemented with 10% FBS and 1% pen/strep at 10% $CO_2$ at 37 °C. RPE1 and U2OS cells were validated by STR sequencing and all cell lines were routinely tested for mycoplasma negativity. For ATF3 validation, cells were incubated in 2 µg/mL doxorubicin hydrochloride (Fisher Sci) for 1 hr prior to fixation. For EGR1 validation, cells were incubated in 5 ng/mL hEGF (Peprotech) for 1 hr prior to fixation. For MN induction, cells were incubated 100 nM BAY1217389 (Msp1i, Fisher Sci) for the indicated times. Where noted, Cdk1i (RO-3306, Sigma) was added to 10 µM.

hTERT RPE-1 NLS-3xDendra2/H2B-emiRFP703 and U2OS NLS-3xDendra2/H2B-emiRFP703 cell lines were produced through serial transduction of lentiviruses. Lentivirus was produced in HEK293T cells using standard protocols and filtered medium was added with polybrene (Sigma, #H9268) for transduction. Cells were selected with 10 µg/mL blasticidin (Invivogen) and 500 µg/mL active G418 (Gibco) and FACS sorted on an Aria II sorter (BD Biosciences) for the top 20% brightest double positive cells. hTERT RPE-1 NLS-3xDendra2-P2A-H2B-miRFP703 cells were created through viral transduction and FACs sorting for the brightest double positive population. MCC13 cells were a gift from Dr. Paul Nghiem (University of Washington). HeLa H2B-GFP cells were a gift from Dr. Daphne Avgousti (Fred Hutchinson Cancer Center) and were originally acquired from Millipore (SCC117). hTERT-HFF cells were a gift from Dr. Denise Galloway (Fred Hutchinson Cancer Center) (*Kiyono et al., 1998*).

## Microscopy

Visual cell sorting experiments were performed on a Leica DMi8 widefield fluorescence microscope with Adaptive Focus outfitted with an i8 incubation chamber (Leica) with temperature (PeCon: Temp-Controller 2000–1) and gas control (Oko) and a Mosaic 3 Digital Micromirror (Andor). Images were acquired with a 20x0.8 NA apochromatic objective (Leica) using an iXon Ultra 888 EMCCD camera (Andor) and MetaMorph v7.1.0.1.161 (Molecular Devices). Camera binning was set to 2 and single section images were acquired. For quality control, the first and last 5 positions were reimaged at the end of each well for in all channels and images of an unactivated well were acquired for comparison.

Fixed cell training, validation, and testing images for VCS MN and MNFinder were acquired with a Leica DMi8 laser scanning confocal microscope using the Leica Application Suite (LAS X) software and a 40 x/1.3 NA Oil APO CS objective (Leica) or 20 x/0.7 NA PLAN APO objective, or a Leica DMi8 microscope outfitted with a Yokogawa CSU spinning disk unit, Andor Borealis illumination, ASI automated Stage with Piezo Z, with an environmental chamber and Automatic Focus using a 40 x/1.3 NA Oil PLAN APO objective. Images on the spinning disk microscope were captured using an iXon Ultra 888 EMCCD camera and MetaMorph software (v7.10.4).

## Micronucleus segmentation and cell classifiers

*VCS MN:* The neural net was created using the FastAI 1.0 library in Python, a UNet with Torchvision's ResNet18 pre-trained model as its base architecture (*Ronneberger et al., 2015*). Training for MN recognition was performed using ~2000 images of individual cells as training data, a further 164 for validation, and 177 for testing. Training images were of RPE-1 NLS-3xDendra2-P2A-H2B-miRFP703 cells after incubation in 0.5 µM reversine (an Mps1 inhibitor, EMD Millipore) or DMSO for 24 hr and taken with a ×20 widefield objective on the visual cell sorting microscope. Nuclei were segmented on H2B channel images using the Deep Retina segmenter (*Caicedo et al., 2019a*) and 48x48 px image crops were generated centered on each nucleus. For training, MN pixels in cropped images were manually annotated by an MN expert. MN associated with chromatin bridges were not annotated to ensure that labeled MN were discrete nuclear compartments.

The VCS MN classifier takes as input a two-channel ×20 image. It applies the Deep Retina segmenter to the H2B channel to segment nuclei, discards any nuclei touching the edge of the image, and generates a 48x48 px crops centered on each nucleus. Each crop is processed with Sobel edge detection and linearly enlarged to 96x96 px. To accommodate the ResNet18 3-channel architecture, each crop is expanded to the H2B channel, a duplicate of the H2B channel, and the results of Sobel edge detection. Identified MN are mapped back to the full image and assigned to the closest segmented nucleus. MN more than 40 px away from a nucleus are discarded.

Once MN are assigned to nuclei, the classifier calculates the maximum Dendra2 MN/nucleus intensity ratio for each MN. MN with a ratio below 0.16 are classified as ruptured. This threshold was identified using the JRip classifier in Weka 3.8.6 to define the optimal threshold to separate manually annotated intact and ruptured MN (*Cohen, 1995*; *Witten et al., 2017*). Nucleus masks are classified as MN +or MN- based on the presence or absence of at least one associated MN. MN +nuclei are then further classified into those associated with only intact MN (rupture-) or those with at least one ruptured MN (rupture+).

For analysis of MN recall and PPV, MN ground truth labels were generated using PixelStudio 4.5 on an iPad (Apple) for annotation. Recall was calculated as the proportion of all MN that overlapped with a predicted segment. Positive predictive value was calculated as the proportion of all predicted segments that overlapped with a MN. Mean Intersection over Union (mIoU) was calculated per object by quantifying the overlap between groups of true positive pixels and their respective ground truths.

Prior to analysis of U2OS cells, the VCS MN segmentation module was retrained on a collection of images of RPE1, U2OS, HFF, and HeLa cells that had been incubated in 100 nM BAY1217389 or 0.5 µM reversine for 24 hr to increase MN frequency. Live images of RPE1 and U2OS NLS-3xDendra2/H2B-emiRFP703 cells were acquired on the VCS microscope at ×20. Images of fixed cells were taken on either the LSM or spinning disk confocal microscopes at ×40 after fixation in 4% paraformaldehyde (Electron Microscopy Sciences, #15710) for 5 min at room temperature. Cells were labeled with DAPI where indicated. ~2300 crops of U2OS NLS-3xDendra2/H2B-emiRFP703 cells taken on the VCS microscope at ×20 were used for training with another 233 held back for validation and 910 for testing. Three images of Hoechst labeled U2OS cells taken at ×20 on a widefield

microscope at 16-bit depth were downloaded from the Broad Bioimage Benchmark Collection (BBBC039v1, *Bray et al., 2016*; *Caicedo et al., 2019b*; *Ljosa et al., 2012*) and linearly scaled up by 0.5 for testing. For all images in the training, validation, and testing datasets, Deep Retina segmentation was bypassed by manually annotating the nuclei and generating image crops with a custom python program. These crops and masks were then fed into VCS MN to obtain MN masks and nucleus/MN assignments.

*MNFinder:* The MNFinder neural nets were created using TensorFlow 2.0 without transfer learning. Training was performed on 128x128 px crops generated from widefield images of RFP703/Dendra RPE-1 and U2OS cells of H2B-RFP703 and confocal images of DAPI-labeled MCC13 and HeLa cells after Mps1i incubation (*Table 3*). Nuclei and MN pixels were annotated by hand by an MN expert.

For nucleus/MN segmentation, predictions are taken from two UNet-based neural nets, with MN predictions fed into a third ensembling UNet. All UNets are identical, save for the incorporation of multiscale downsampling into one of the input UNets. For "cell" segmentation (instance segmentation) a UNet architecture incorporating 3 decoder pathways is used to predict distance maps, proximity maps, and foreground pixels. The distance and proximity map decoders incorporate features from a UNet3 +design: specifically, additional skip connections from multiple layers of the encoder and decoder pathways and deep supervision during training (*Huang et al., 2020*). Training data were generated from annotated nuclei and MN images by generating a concave hull grouping a nucleus and associated MN using the cdBoundary package in Python (*Duckham et al., 2008*). This hull was transformed into a distance map by calculating the Euclidean distance transform (EDT) with each pixel value encoding the shortest distance between that pixel and the background. Proximity maps were generated by setting all pixels as foreground pixels except for those belonging to other hulls and applying an EDT, masked by the cell's boundaries, and raising this to the 4th power to sharpen edges. Both maps are scaled from 0 to 1 for each cell.

MNFinder requires images with an image resolution of 1.55–2.8 px/μm, or nuclei around 30 px in diameter. Images taken at other resolutions need to be linearly scaled to within this range prior to analysis. Both single channel chromatin and multi-channel images can be used as input. MNFinder processing proceeds by first cropping input images using a 128x128 px sliding window, advancing the window by 96 px horizontally and vertically to oversample the image. Crops are then expanded into two-channel images, with the second channel the result of Sobel edge detection. These images are processed by the neural nets, post-processed as described, and reassembled by linear blending into a complete field. Recall and positive predictive values were calculated using the same metrics as for VCS classifier validation.

MNFinder was validated on images of RPE1, U2OS, HFF, and HeLa cells described in *Table 3*. PPV and recall for MN segmentation were calculated for individual input UNets and the ensemble UNet (*Table 1*).

## Outline of visual cell sorting cell isolation experiments

Cells for visual cell sorting were plated onto six-well glass-bottom, black-walled plates (CellVis) at a density of 50,000–225,000 cells per well 1–2 days before activation. An extra unactivated well and a well of non-fluorescent RPE1 cells were plated as controls. One day before imaging, 100 nM Mps1i was added to the medium. One hour prior to imaging, cells were washed 1 x in PBS and medium changed to phenol red free (GIBCO) containing 10 μM RO-3306 (Sigma). The plate was transferred to the microscope, the plate center and micromirror device were aligned, and the appropriate journals (see *Hasle et al., 2020*) were initiated for visual cell sorting. Imaging conditions were optimized for each experiment. Images were acquired using MetaMorph and analyzed on a dedicated linked computer. 1-bit masks of MN +nuclei and MN- nuclei were transmitted back to MetaMorph, which directed UV pulses at the segmented nuclei. Activation occurred using either a 200ms or 800ms pulse of the 405 nm laser. Classifier predictions were compared to the first 3 and last 3 images from each VCS experiment, each manually annotated prior to downstream analysis, including RNA extraction.

Activated and unactivated cells were trypsinized (0.25% Trypsin-EDTA, GIBCO), suspended in 2% FBS in PBS, and sorted using a FACS Aria II (BD Biosciences). Compensation for PE-blue excitation of unconverted Dendra2 was performed on the unactivated cells. Dendra2 activation-based sorting gates were defined on single cells positive for both Dendra2 and emiRFP703 using the PE-Blue-A/FITC-A ratio, which was found to be more distinct than the PE-YG/FITC ratios. Cells were sorted

into 2% FBS then pelleted and either flash frozen on dry ice or replated onto poly-L-lysine coated coverslips.

## CellTrace

Activation and sorting accuracy were analyzed for RPE1 RFP703/Dendra2 cells by incubating cells in CellTrace far-red (Thermo Fisher) for 10 min at 37 °C, trypsinizing and pelleting cells, mixing 1:1 with unlabeled cells, and plating. A classifier segmented nuclei with the Deep Retina segmenter on Dendra (green) fluorescence and measured the mean far-red intensity in the nucleus (*Hasle et al., 2020*). Threshold intensity for activation was experimentally determined. Cells were sorted by FACs for Dendra2 ratio and CellTrace intensity using compensation to eliminate emiRFP703 spectral overlap and then reanalyzed on the same machine.

## Mps1i+/-isolation

Cells incubated in Msp1i or DMSO were imaged and activated using a random classifier. 1-bit masks of nuclei generated using the Deep Retina segmenter were randomly assigned to receive 800ms or 200ms pulses. At least 13 k cells were collected per sorting bin and samples were pelleted, flash frozen, and stored at –80 °C.

## Micronucleus+/- isolation

Two wells were imaged sequentially per experiment with the activation time for MN+ and MN- nuclei reversed between wells.

## Rupture +/-isolation

Cells were plated 2 days before imaging in medium containing 1 µM Cdk4/6i (PD-0332991, Sigma). Twenty-four hours later, cells were rinsed 3 x with PBS and the medium replaced with 100 nM BAY1217389. Only one well was imaged per experiment with rupture- cells receiving 800ms and rupture + cells receiving 200ms pulses.

## RNA isolation and sequencing

We extracted RNA from frozen cell pellets using the RNAqueous micro kit (Thermo Fisher), according to the manufacturer's protocol. Residual DNA was removed by DNase I treatment and RNA was further purified by glycogen precipitation (RNA-grade glycogen; Thermo Fisher) and resuspension in ultra-pure $H_2O$ heated to 65 °C. RNA quality and concentration was checked by the Genomics Core at the Fred Hutchinson Cancer Center with an Agilent 4200 Tapestation HighSense RNA assay and only samples with RIN scores above 8 and 28 S/18 S values above 2 were further processed. cDNA synthesis and library preparations were performed by the Genomics Core using the SMARTv4 for ultra-low RNA input and Nextera XT kits (Takara). Sequencing was also performed on an Illumina NextSeq 2000 sequencing system with paired-end, 50 bp reads.

## RNAseq and gene-set enrichment analysis

We quantified transcripts with Salmon to map reads against the UCSC hg38 assembly at http://refgenomes.databio.org (digest: 2230c535660fb4774114bfa966a62f823fdb6d21acf138d4), using bootstrapped abundance estimates and corrections for GC bias (*Patro et al., 2017*). For comparisons with data from He, et al. and Santaguida, et al., the original FASTA files deposited at the Sequence Read Archive were downloaded with NCBI's SRA Toolkit and quantified with Salmon (*He et al., 2019*; *Santaguida et al., 2017*). No GC-bias correction was applied as only single-end reads were available.

Transcript abundances were processed to find differentially expressed genes (DEGs) with the R package DESeq2 version 3.16 in R 4.2.1, RStudio 2022.07.2 build 576, and Sublime Text build 4143. Files were imported into DESeq2 with the R package tximeta (*Love et al., 2020*; *Love et al., 2014*), estimated transcript counts were summarized to gene-level, and low-abundance genes were filtered by keeping only those genes with estimated counts ≥700 in at least two samples. DEGs were identified using a likelihood ratio test comparing the full model with one with the condition of interest dropped and an FDR of 0.05. Log-fold changes were corrected using empirical Bayes adaptive shrinkage (*Stephens, 2017*). Operations were performed before pseudogenes were filtered from dataset.

GSEA was performed using the R package fgsea version 1.25.1, comparing log-fold changes of all DEGs against the full *Homo sapiens* Hallmark Gene Sets version 2022.1, part of the MSigDB resource (UC San Diego, Broad Institute) (*Crameri, 2018*; *Greene et al., 2017*; *Korotkevich et al., 2021*; *Liberzon et al., 2015*; *Subramanian et al., 2005*).

## Live-cell imaging for MN rupture frequency analysis

RPE1 NLS-3xDendra2/H2B-emiRFP703 cells were plated 2 days before imaging and treated for 24 hr with either 1 μM Cdk4/6i or DMSO. One day before imaging, cells were rinsed and incubated in 100 nM BAY1217389. Nineteen hours later, the media was exchanged for Cdk1i medium, five positions were imaged in each well and rupture- cells were activated. These positions and the surrounding area were imaged every hour for 11 hr and the status of photoconverted cells manually recorded.

## Immunofluorescence (IF)

Cells plated on poly-L-lysine coated coverslips or glass bottomed plates were fixed for IF in 4% paraformaldehyde for 5 min unless otherwise indicated. Cells were permeabilized for 30 min at RT in PBSBT (1xPBS (GIBCO), 3% BSA, 0.4% Triton X-100, 0.02% sodium azide (all Sigma)), followed by incubation in primary antibodies diluted in PBSBT for 30 min, secondary antibodies diluted in same for 30 min, and 5 min in 1 μg/mL DAPI (Invitrogen). Coverslips were mounted in VectaShield (VectorLabs) and sealed with nail polish before imaging. Primary antibodies used were: mouse-a-$\gamma$H2AX-phospho Ser139 (1:500; BioLegend, 613401), rabbit-a-ATF3 (1:400; Cell Signaling Technology, 18665), rabbit-a-EGR1 (1:1600; Cell Signaling Technology, 4154), and rabbit α H3K27Ac (1:500; Abcam, ab4729). Secondary antibodies used were: AF647 goat-a-mouse (1:1000; Life Technologies, A21236) and AF488 goat-a-rabbit (1:2000; Life Technologies, A11034).

## DNA FISH

RPE1 cells plated onto poly-L lysine coverslips were fixed in –20 °C 100% methanol for 10 min, rehydrated for 10 min in 1xPBS and processed for IF. Cells were then refixed in 4% PFA for 5 min at RT then incubated in 2xSSC (Sigma) for 2x5 min RT. Cells were permeabilized in 0.2 M HCl (Sigma), 0.7% TritonX-100 in $H_2O$ for 15 min at RT, washed in 2xSSC, and incubated for 1 hr at RT in 50% formamide (Millipore). Cells were rewashed in 2xSSC, inverted onto chr 1, 11, or 18 XCE probes (MetaSystems), and the coverslips sealed with rubber cement. Probes were hybridized at 74 °C for 3 min and then incubated for 4 hr (chr 18) or overnight (chr.s 1 and 11) at 37 °C. After hybridization, coverslips were washed in 0.4xSSC at 74 °C for 5 min, then 2xSCC 0.1% Tween20 (Fisher) for 2x5 min at RT. DNA was labeled by incubation in 1 μg/mL DAPI for 5 min at RT, and coverslips mounted in VectaShield. Images were acquired as 0.45 μm step z-stacks through the cell on the confocal LSM with a ×40 objective. Cells that had more or less than two FISH foci were classified as aneuploid for that chromosome.

For MN chromosome enumeration, Mps1i treated RPE1 cells were labeled with a PNA CENPB-Cy5 probe (PNA Bio). The same DNA FISH protocol was followed until formamide denaturation. At that point cells were incubated in 50% formamide in 2 x SSC for 30 min at 85 °C then rinsed three times in ice cold 2xSSC. PNA probes were diluted to 50 μM in 85 °C hybridization buffer (60% formamide +20 mM Tris, pH 7.4+0.1 μg/mL salmon sperm DNA (Trevigen)) and coverslips were simultaneously washed at 85 °C in 2xSSC. Coverslips were then incubated in 10 μL of the PNA probe for 10 min at 85 °C and then 2 hr at RT. Coverslips were then washed twice with 2 x SSC +0.1% Tween-20 at 55 °C for 10 min and once with 2 x SSC +0.1% Tween-20 at RT. before incubation in DAPI and mounting in Prolong Gold (Life Technologies).

## Image analysis
### Mps1i nuclear morphology

Analysis was performed on images plus ground truth annotations and Deep Retina masks acquired as part of the DMSO versus Mps1i visual cell sorting RNAseq experiment. Solidity was quantified on ground truth and Deep Retina nuclear masks and chromatin bridges, misshapen nuclei, and dead or mitotic nuclei were manually scored. Nuclear atypia was defined as a cell with multiple large nuclei (rare) or a nuclear shape that multiple significant indentations. Nuclei that curved around a single focus were scored as normal.

## Cell confluency

A subset of images of RFP703/Dendra cells from the MN +/-RNASeq experiments were thresholded on the 3xDendra-NLS signal at a value that included the cytoplasmic signal and the proportion of thresholded pixels was calculated per image using ImageJ. Distances between nuclei borders were calculated using ground truth masks of the same images using tools packaged in the 3D ImageJ Suite (*Ollion et al., 2013*).

## Dendra2 ratio stability

Nuclei were segmented on images taken at the start and end of an Mps1i+/-VCS experiment by thresholding on the GFP channel, measuring the mean intensity of GFP and RFP, and calculating the RFP:GFP ratio per nucleus for each image group.

## MN +/-sorting accuracy

Cells replated and fixed after sorting were imaged on the LSM confocal at 40 x with 0.45 μm z-stacks through the cell. Image names were randomized prior to quantification of MN + cells.

## ATF3 and EGR1 intensity

Images were acquired as 0.45 μm step z-stacks through the cell on the confocal LSM with a ×40 objective. Images were corrected for illumination inhomogeneity by dividing by a dark image and background subtracted using a 60 px radius rolling-ball in FIJI (*Schindelin et al., 2012*) (v2.9.0). Single in focus sections of each nucleus was selected and nuclei masks generated by thresholding on RFP-NLS. Mean intensity of ATF3 or EGR1 were calculated for each nucleus and normalized for each replicate by scaling to the median value for the DMSO control. Statistics were calculated on the raw values.

## Statistical analyses

Shorthand p-values are as follows:

> ns: p-value ≥ 0.05
> *: p-value <0.05
> **: p-value <0.01
> ***: p-value <0.001
> ****: p-value <0.0001

Generalized estimating equations (GEE) were used to determine statistical differences for nominal data with multiple variables using binomial distributions and a logit link function (*Halekoh et al., 2006*). For *Figure 1c*, data were assessed using the formula: (*# recalled, # missed)~MN status* where *MN status* is whether ruptured or intact. For *Figure 6c*, we also used a binomial distribution and a logit link function. For *Figure 5c*, *Figure 6—figure supplement 1c*, we used the formula: (*# aneuploid, # normal)~Status ×* Chr where *Status* is whether the cell was MN +/-or Rupture +/-and *Chr* is chromosome identity. p-values for each individual property were calculated using the drop1 function in R. In *Figure 6c*, *Figure 6—figure supplement 1a-b*, we used a gamma distribution and the formula *mean intensity ~Population*. Statistical significance for differences between single nominal variables in other figures were by Barnard's exact test. KS test=Kolmogorov-Smirnov test to compare distributions of non-Gaussian data.

The predicted change to classifier PPV in *Figure 5—figure supplement 1c* was determined by reducing the true positive rate in the rupture- population by the difference in mean rupture frequencies between the beginning and end of the experiments and increasing the true positive rate in the rupture +population by the same.

## Acknowledgements

This work was supported by a National Institutes of Health grant (R35GM124766, awarded to EM Hatch), a National Human Genome Research Institute grant (RM1HG010461, awarded to DM Fowler), a training grant (T32CA009657, awarded to L DiPeso), the Rita Allen Foundation Scholars program (awarded to EM Hatch), and the Fred Hutchinson Cancer Center Bioinformatics and Genomics cores (funded by NIH grant P30CA015704).

## Additional information

### Funding

| Funder | Grant reference number | Author |
|---|---|---|
| National Institute of General Medical Sciences | R35GM124766 | Lucian DiPeso<br>Heather Z Huang<br>Emily M Hatch |
| National Human Genome Research Institute | RM1HG010461 | Sriram Pendyala<br>Douglas M Fowler |
| National Cancer Institute | T32CA009657 | Lucian DiPeso |
| Rita Allen Foundation | Scholars Program | Emily M Hatch |
| National Cancer Institute | P30CA015704 | Lucian DiPeso<br>Heather Z Huang<br>Emily M Hatch |

The funders had no role in study design, data collection and interpretation, or the decision to submit the work for publication.

### Author contributions

Lucian DiPeso, Conceptualization, Software, Formal analysis, Validation, Investigation, Visualization, Methodology, Writing - original draft; Sriram Pendyala, Software, Formal analysis, Methodology; Heather Z Huang, Formal analysis, Methodology; Douglas M Fowler, Conceptualization, Supervision, Funding acquisition, Project administration; Emily M Hatch, Conceptualization, Supervision, Funding acquisition, Project administration, Writing – review and editing

### Author ORCIDs

Douglas M Fowler ⬡ https://orcid.org/0000-0001-7614-1713
Emily M Hatch ⬡ https://orcid.org/0000-0002-3393-4075

Reviewer #1 (Public review): https://doi.org/10.7554/eLife.101579.3.sa1
Reviewer #2 (Public review): https://doi.org/10.7554/eLife.101579.3.sa2
Reviewer #3 (Public review): https://doi.org/10.7554/eLife.101579.3.sa3
Author response https://doi.org/10.7554/eLife.101579.3.sa4

---

## Additional files

### Supplementary files

Supplementary file 1. Additional tables containing transcriptomics data. (a) DMSO 800 vs 200 DEG. List of differentially expressed genes (FDR ≤0.05) in DMSO treated RPE1 cells isolated after an 800ms or 200ms UV pulse. For all DEG analyses: padj = false-discovery rate adjusted p-values. (b) *Table 2*: Msp1i DEG. List of differentially expressed genes (FDR ≤0.05) in Mps1i versus DMSO RPE1 cells. (c) Msp1i DEG $\log_2$FC. List of differentially expressed genes in Mps1i versus DMSO RPE1 cells filtered for $\log_2$ fold change above 0.58 or below –0.58. (d) He et al DEG. List of differentially expressed genes (FDR ≤0.05) in RPE1 cells treated with nocodazole for 8 h versus control, initially reported in *He et al., 2018*. Overlap with Mps1i DEG list noted in last column. (e) Santaguida et al DEG. List of differentially expressed genes (FDR ≤0.05) in RPE1 cells that were treated with the Mps1i molecule reversine for 12 h, released, and ceased to divide, compared to control cells. Results initially reported in *Santaguida et al., 2017*. Overlap with Mps1i DEG list noted in last column. (f) Hallmark Mps1i. List of MSigDB Hallmark gene sets that are significantly enriched (FDR ≤0.05) in Mps1i treated RPE1 cells. For all Hallmark lists, NES=normalized enrichment score, padj = false discovery rate adjusted p-values, size = number of genes in gene set, leading_edge = , sig_genes_in_geneset=the gene names of set genes that were significantly upregulated. (g) Hallmark He et al. List of MSigDB Hallmark gene sets that are significantly enriched in nocodazole treated RPE1 cells from *He et al., 2018*. (h) Hallmark Santaguida et al. List of MSigDB Hallmark gene sets that are significantly enriched in reversine treated RPE1 cells from *Santaguida et al., 2017*. (i) MN +DEG. List of differentially expressed genes (FDR ≤0.05) in Mps1i treated RPE1 cells

with MN versus without MN. (j) MN +DEG log2FC. List of differentially expressed genes in Mps1i treated RPE1 cells with MN versus without MN filtered for $\log_2$ fold change above 0.58 or below –0.58. Overlap with Mps1i DEG list noted in last column. (k) Ruptured +DEG. List of differentially expressed genes (FDR ≤0.05) in Mps1i treated micronucleated RPE1 cells with only intact MN versus at least 1 ruptured MN. (l) Ruptured DEG +log2 FC. List of differentially expressed genes in Mps1i treated micronucleated RPE1 cells with only intact MN versus at least 1 ruptured MN MN filtered for absolute $\log_2$ fold change ≥0.58. Overlap with Mps1i DEG list noted in last column. (m) Hallmark MN+. List of MSigDB Hallmark gene sets that are significantly enriched in Mps1i treated RPE1 cells with MN. (n) Hallmark ruptured+. List of MSigDB Hallmark gene sets that are significantly enriched in Mps1i treated RPE1 cells with MN and at least 1 ruptured MN versus no ruptured MN. (o) Log$_2$FC per replicate. List of $\log_2$ fold change values for Mps1i DEGs with absolute $\log_2$FC ≥0.58 broken out by MN +and rupture +replicate. Genes with all NA values for either all MN +or all rupture +replicates were excluded from analysis. enriched_in_rupture+_cluster=genes present in cluster enriched in increased expression over Mps1i DEGs.

MDAR checklist

## Data availability

All cell lines and plasmids made for this study are available upon request. Both the MNFinder and VCS MN modules and supporting code are available on the Hatch Lab GitHub repository, along with all training, testing, and validation images and their ground truth labels (https://github.com/hatch-lab/mnfinder and https://github.com/hatch-lab/fast-mn). Additional validation images and data are available on BioImage Archive. Raw and processed RNA-seq datasets were deposited in the National Center for Biotechnology Information (NCBI) Gene Expression Omnibus (GEO), accession number GSE291811.

The following datasets were generated:

| Author(s) | Year | Dataset title | Dataset URL | Database and Identifier |
|---|---|---|---|---|
| DiPeso L, Pendyala S, Huang HZ, Fowler DM, Hatch EM | 2025 | Image-based identification and isolation of micronucleated cells to dissect cellular consequences | https://www.ncbi.nlm.nih.gov/geo/query/acc.cgi?acc=GSE291811 | NCBI Gene Expression Omnibus, GSE291811 |
| Hatch E | 2025 | Image-based identification and isolation of micronucleated cells to dissect cellular consequences | https://www.ebi.ac.uk/biostudies/bioimages/studies/S-BIAD1990 | BioImage Archive, S-BIAD1990 |

The following previously published datasets were used:

| Author(s) | Year | Dataset title | Dataset URL | Database and Identifier |
|---|---|---|---|---|
| Santaguida S, Richardson A, Iyer D, M'Saad O, Zasadil L, Knouse K, Wong Y, Rhind N, Desai A, Amon A | 2017 | Aneuploidy triggers an immune response | https://www.ncbi.nlm.nih.gov/geo/query/acc.cgi?acc=GSE83647 | NCBI Gene Expression Omnibus, GSE83647 |
| He Q, Au B, Kulkarni M, Shen Y, KahJ L, Maimaiti J, Chong HC, Lim EH, Rancati G, Sinha I, Fu Z, Wang X, Crasta KC, ChengKit W, MoniqueNH L, JohnE C | 2019 | Genome-wide transcriptional response to random aneuploidy in human cells | https://www.ncbi.nlm.nih.gov/geo/query/acc.cgi?acc=GSE109519 | NCBI Gene Expression Omnibus, GSE109519 |

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
