## [Editor Report · eLife Assessment]

This **valuable** paper reports machine learning-based image analysis pipelines for the automated segmentation of micronuclei and the detection and sorting of micronuclei-containing cells. These are powerful new tools for researchers who study micronuclei and their physiologic consequences. The analysis of the new tools and their benchmarking is rigorous and **convincing**; applications and remaining limitations are well explained in the paper.

---

## [Referee Report · Reviewer #1 (Public review)]

DiPeso et al. develop two tools to (i) classify micronucleated (MN) cells, which they call VCS MN, and (ii) segment micronuclei and nuclei with MNFinder. They then use these tools to identify transcriptional changes in MN cells.

The strengths of this study are:

- Developing highly specialized tools to speed up the analysis of specific cellular phenomena such as MN formation and rupture is likely valuable to the community and neglected by developers of more generalist methods.

- A lot of work and ideas have gone into this manuscript. It is clearly a valuable contribution.

- Combining automated analysis, single-cell labeling, and cell sorting is an exciting approach to enrich for phenotypes of interest, which the authors demonstrate here.

The authors addressed my original concerns related to the first version of this manuscript.

---

## [Referee Report · Reviewer #2 (Public review)]

Summary:

Micronuclei are aberrant nuclear structures frequently seen following the missegregation of chromosomes. The authors present two image analysis methods, one robust and another rapid, to identify micronuclei (MN) bearing cells. To analyse their software efficacy, the authors study images of cells treated with MPS1 inhibitor to induce chromosome missegregation. Next, the authors use RNA-seq to assess the outcomes of their MN-identifying methods: they do not observe a transcriptomic signature specific to MN but find changes that correlate with aneuploidy status. Overall, this work offers new tools to identify MN-presenting cells, and it sets the stage with clear benchmarks for further software development.

Strengths:

Currently, there are no robust MN classifiers with a clear quantification of their efficiency across cell lines (mIoU score). The software presented here tries to address this gap. GitHub material (images, ground truth labels, tools, protocols, etc.) provided is a great asset to computational biologists. The method has been tested in more than one cell line. This method can help integrate cell biology and 'omics' data, making it suitable for multimodal studies.

Weaknesses:

Although the classifier outperforms available tools for MN segmentation by providing mIoU, it's not yet at a point where it can be reliably applied to functional genomics assays where we expect a range of phenotypic penetrance in most cell lines (e.g., misshapen, multinucleated, and lagging DNA in addition to micronucleated cells). The discussion considers the nature and proportion of MN in RPE1 cells, and how the classifier is well-suited for RPE1 that predominantly display MN structures. Whether the classifier can rigorously assign MN-presenting cells amidst drastic nuclear aberrancies following a spindle checkpoint loss needs to be tested in the future.

---

## [Referee Report · Reviewer #3 (Public review)]

Summary:

The authors develop automated methods to visually identify micronuclei (MN) and MN-containing cells. The authors then use these methods to isolate MN-containing RPE-1 cells post-photoactivation and analyze transcriptional changes in cells with and without micronuclei. The authors find that RPE-1 cells with MN have similar transcriptomic changes as aneuploid cells and that MN rupture does not lead to vast changes in the transcriptome.

Strengths:

The authors develop a method that allows for automating measurements and analysis of micronuclei. This has been something that the field has been missing for a long time. Using such a method has the potential to greatly enhance the field's ability to analyze micronuclei and understand the downstream consequences. The authors also develop a method to identify cells with micronuclei in real-time, mark them using photoconversion, and then isolate them via cell sorting, which could change the way we isolate and study MN-containing cells, and the scale at which we do it. The authors use this method to look at the transcriptome. This method is very powerful as it can allow for the separation of a heterogenous population and subsequent analysis with a much higher sample number than previously possible.

Weaknesses:

The major weakness of this paper is the transcriptomic analysis of MN. There is in general large variance between replicates in experiments looking at cells with ruptured versus intact micronuclei. This limits our ability to assess if lack of changes are due to truly not having changes between these populations or experimental limitations. More transcriptomic analysis will be necessary to fully understand the downstream consequences of MN rupture.

---

## [Author Response]

The following is the authors’ response to the original reviews.

**Public Reviews:**

**Reviewer #1 (Public review):**
DiPeso et al. develop two tools to (i) classify micronucleated (MN) cells, which they call VCS MN, and (ii) segment micronuclei and nuclei with MMFinder. They then use these tools to identify transcriptional changes in MN cells.The strengths of this study are:(1) Developing highly specialized tools to speed up the analysis of specific cellular phenomena such as MN formation and rupture is likely valuable to the community and neglected by developers of more generalist methods.(2) A lot of work and ideas have gone into this manuscript. It is clearly a valuable contribution.(3) Combining automated analysis, single-cell labeling, and cell sorting is an exciting approach to enrich phenotypes of interest, which the authors demonstrate here.Weaknesses:(1) Images and ground truth labels are not shared for others to develop potentially better analysis methods.

We regret this omission and thank the reviewer for pointing it out. Both the images and ground truth labels for VCS MN and MNFinder are now available on the lab’s github page and described in the README.txt files. VCS MN: https://github.com/hatch-lab/fast-mn. MNFinder: https://github.com/hatch-lab/mnfinder.

(2) Evaluations of the methods are often not fully explained in the text.

The text has been extensively updated to include a full description of the methods and choices made to develop the VCS MN and MNFinder image segmentation modules.

(3) To my mind, the various metrics used to evaluate VCS MN reveal it not to be terribly reliable. Recall and PPV hover in the 70-80% range except for the PPV for MN+. It is what it is - but do the authors think one has to spend time manually correcting the output or do they suggest one uses it as is?

VCS MN attempts to balance precision and recall with speed to reduce the fraction of MN changing state from intact to ruptured during a single cell cycle during a live-cell isolation experiment. In addition, we chose to prioritize inclusion of small MN adjacent to the nucleus in our positive calls. This meant that there were more false positives (lower PPV) than obtained by other methods but allowed us to include this highly biologically relevant class of MN in our MN+ population. Thus, for a comprehensive understanding of the consequences of MN formation and rupture, we recommend using the finder as is. However, for other visual cell sorting applications where a small number of highly pure MN positive and negative cells is preferred, such as clonal outgrowth or metastasis assays, we would recommend using the slower, but more precise, MNFinder to get a higher precision at a cost of temporal resolution. In addition, MNFinder, with its higher flexibility and object coverage, is recommended for all fixed cell analyses.

**Reviewer #2 (Public review):**
Summary:Micronuclei are aberrant nuclear structures frequently seen following the missegregation of chromosomes. The authors present two image analysis methods, one robust and another rapid, to identify micronuclei (MN) bearing cells. The authors induce chromosome missegregation using an MPS1 inhibitor to check their software outcomes. In missegregation-induced cells, the authors do not distinguish cells that have MN from those that have MN with additional segregation defects. The authors use RNAseq to assess the outcomes of their MN-identifying methods: they do not observe a transcriptomic signature specific to MN but find changes that correlate with aneuploidy status. Overall, this work offers new tools to identify MN-presenting cells, and it sets the stage with clear benchmarks for further software development.Strengths:Currently, there are no robust MN classifiers with a clear quantification of their efficiency across cell lines (mIoU score). The software presented here tries to address this gap. GitHub material (tools, protocols, etc) provided is a great asset to naive and experienced computational biologists. The method has been tested in more than one cell line. This method can help integrate cell biology and 'omics' studies.Weaknesses:Although the classifier outperforms available tools for MN segmentation by providing mIOU, it's not yet at a point where it can be reliably applied to functional genomics assays where we expect a range of phenotypic penetrance.

We agree that the MNFinder module has limitations with regards to the degree of nuclear atypia and cell density that can be tolerated. Based on the recall and PPV values and their consistency across the majority conditions analyzed, we believe that MNFinder can provide reliable results for MN frequency, integrity, shape, and label characteristics in a functional genomics assay in many commonly used adherent cell lines. We also added a discussion of caveats for these analyses, including the facts that highly lobulated nuclei will have higher false positive rates and that high cell confluency may require additional markers to ensure highly accurate assignment of MN to nuclei.

Spindle checkpoint loss (e.g., MPS1 inhibition) is expected to cause a variety of nuclear atypia: misshapen, multinucleated, and micronucleated cells. It may be difficult to obtain a pure MN population following MPS1 inhibitor treatment, as many cells are likely to present MN among multinucleated or misshapen nuclear compartments. Given this situation, the transcriptomic impact of MN is unlikely to be retrieved using this experimental design, but this does not negate the significance of the work. The discussion will have to consider the nature, origin, and proportion of MN/rupture-only states - for example, lagging chromatids and unaligned chromosomes can result in different states of micronuclei and also distinct cell fates.

We appreciate the reviewer’s comments and now quantify the frequency of other nuclear atypias and MN chromosome content in RPE1 cells after 24 h Mps1 inhibition (Fig. S1). In summary, we find only small increases in nuclear atypia, including multinucleate cells, misshapen nuclei, and chromatin bridges, compared to the large increase in MN formation. This contrasts with what is observed when mitosis is delayed using nocodazole or CENPE inhibitors where nuclear atypia is much more frequent. Importantly, after Mps1 inhibition, RPE1 cells with MN were only slightly more likely to have a misshapen nucleus compared to cells without MN (Fig. S1C).

Interestingly, this analysis showed that the VCS MN pipeline, which uses the Deep Retina segmenter to identify nuclei, has a strong bias against lobulated nuclei and frequently fails to find them (Fig. S2B). Therefore, the cell populations analyzed by RNAseq were largely depleted of highly misshapen nuclei and differences in nuclear atypia frequency between MN+ and MN- cells in the starting population were lost (Fig. S9A, compare to Fig. S1C). This strongly suggests that the transcript changes we observed reflect differences in MN frequency and aneuploidy rather than differences in nuclei morphology.

We agree with the reviewer that MN rupture frequency and formation, and downstream effects on cell proliferation and DNA damage, are sensitive to the source of the missegregated chromatin. In the revised manuscript we make clear that we chose Mps1 inhibition because it is strongly biased towards whole chromosome MN (Fig. S1E), limiting signal from DNA damage products, including chromosome fragments and chromatin bridges. This provides a base line to disambiguate the consequences of micronucleation and DNA damage in more complex chromosome missegregation processes, such as DNA replication disruption and irradiation.

**Reviewer #3 (Public review):**
Summary:The authors develop a method to visually analyze micronuclei using automated methods. The authors then use these methods to isolate MN post-photoactivation and analyze transcriptional changes in cells with and without micronuclei of RPE-1 cells. The authors observe in RPE-1 cells that MN-containing cells show similar transcriptomic changes as aneuploidy, and that MN rupture does not lead to vast changes in the transcriptome.Strengths:The authors develop a method that allows for automating measurements and analysis of micronuclei. This has been something that the field has been missing for a long time. Using such a method has the potential to advance micronuclei biology. The authors also develop a method to identify cells with micronuclei in real time and mark them using photoconversion and then isolate them via FACS. The authors use this method to study the transcriptome. This method is very powerful as it allows for the sorting of a heterogenous population and subsequent analysis with a much higher sample number than could be previously done.Weaknesses:The major weakness of this paper is that the results from the RNA-seq analysis are difficult to interpret as very few changes are found to begin with between cells with MN and cells without. The authors have to use a 1.5-fold cut-off to detect any changes in general. This is most likely due to the sequencing read depth used by the authors. Moreover, there are large variances between replicates in experiments looking at cells with ruptured versus intact micronuclei. This limits our ability to assess if the lack of changes is due to truly not having changes between these populations or experimental limitations. Moreover, the authors use RPE-1 cells which lack cGAS, which may contribute to the lack of changes observed. Thus, it is possible that these results are not consistent with what would occur in primary tissues or just in general in cells with a proficient cGAS/STING pathway.

We agree with the reviewer’s assessment of the limitations of our RNA-Seq analysis. After additional analysis, we propose an alternative explanation for the lower expression changes we observe in the MN+ and Mps1 inhibitor RNA-Seq experiments. In summary, we find that VCS MN has a strong bias against highly lobulated nuclei that depletes this class of cells from both the bulk analysis and the micronucleated cell populations (Fig. S9A). Based on this result, we propose that our analysis reduces the contribution of nuclear atypia to these transcriptional changes and that nuclear morphology changes are likely a signaling trigger associated with aneuploidy.

We believe that this finding strengthens our overall conclusion that MN formation and rupture do not cause transcriptional changes, as suppressing the signaling associated with nuclei atypia should increase sensitivity to changes from the MN. However, we cannot completely rule out that MN formation or rupture cause a broad low-level change in transcription that is obscured by other signals in the dataset.

As to cGAS signaling, several follow up papers and even the initial studies from the Greenburg lab show that MN rupture does not activate cGAS and does not cause cGAS/STING-dependent signaling in the first cell cycle (see citations and discussion in text). Therefore, we expect the absence of cGAS in RPE1 cells will have no effect in the first cell cycle, but could alter the transcriptional profile after mitosis. Although analysis of RPE1 cGAS+ cells or primary cells in these experiments will be required to definitively address this point, we believe that our interpretation of our RNAseq results is sufficiently backed up by the literature to warrant our conclusion that MN formation and rupture do not induce a transcriptional response in the first cell cycle.

**Reviewer #1 (Recommendations for the authors):**
I do not recommend additional experimental or computational work. Instead, I just recommend adapting the claims of the manuscript to what has been done. I am just asking for further clarification and minor rewriting.(1) The manuscript is written like a molecular biology paper with sparse explanations of the authors' reasoning, especially in the development of their algorithms. I was often lost as to why they did things in one way or another.

The revised manuscript has thorough explanations and additional data and graphics defining how and why the VCS MN and MNFinder modules were developed. We hope that this clears up many of the questions the reviewer had and appreciate their guidance on making it more readable for scientists from different backgrounds.

(2) Evaluations of their method are often not fully explained, for example:"On average, 75% of nuclei per field were correctly segmented and cropped.""MN segments were then assigned to 'parent' nuclei by proximity, which correctly associated 97% of MN."Were there ground truth images and labels created? How many? For example, I don't know how the authors could even establish a ground-truth for associating MNs to nuclei if MNs happened to be almost equidistant between two nuclei in their images.I suggest a separate subsection early in the Results section where the underlying imaging data + labels are presented.

We added new sections to the text and figures at the beginning of the VCS MN and MNFinder subsections (Fig. S2 and Fig. S5) with specific information about how ground truth images and labels were generated for both modules and how these were broken up for training, validation, and testing.

We also added information and images to explain how ground truth MN/nucleus associations were derived. In summary, we took advantage of the fact that 2xDendra-NLS is present at low levels in the cytoplasm to identify cell boundaries. This combined with a subconfluent cell population allowed us to unambiguously group MN and nuclei for 98% of MN, we estimate. These identifications were used to generate ground truth labels and analyze how well proximity defines MN/nuclei groups (Fig.s S1 and S2).

(3) Overall, I find the sections long and more subtitles would help me better navigate the manuscript.

Where possible, we have added subtitles.

(4) Everything following "To train the model, H2B channel images were passed to a Deep Retina neural net ..." is fully automated, it seems to me. Thus, there seems to be no human intervention to correct the output before it is used to train the neural network. Therefore, I do not understand why a neural network was trained at all if the pipeline for creating ground truth labels worked fully automatically. At least, the explanations are insufficient.

We apologize for the initial lack of clarity in the text and included additional details in the revision. We used the Deep Retina segmenter to crop the raw images to areas around individual nuclei to accelerate ground truth labeling of MN. A trained user went through each nucleus crop and manually labeled pixels belonging to MN to generate the ground truth dataset for training, validation, and imaging in VCS MN (Fig. S2A).

(5) To my mind, the various metrics used to evaluate VCS MN reveal it not to be terribly reliable. Recall and PPV hover in the 70-80% range except for the PPV for MN+. It is what it is - but do the authors think one has to spend time manually correcting the output or do they suggest one uses it as is? I understand that for bulk transcriptomics, enrichment may be sufficient but for many other questions, where the wrong cell type could contaminate the population, it is not.Remarks in the Results section on what the various accuracies mean for different applications would be good (so one does not need to wait for the Discussion section).

One of the strengths of the visual cell sorting system is that any image analysis pipeline can be used with it. We used VCS MN for the transcriptomics experiment, but for other applications a user could run visual cell sorting in conjunction with MNFinder for increased purity while maintaining a reasonable recall or use a pre-existing MN segmentation program that gives 100% purity but captures only a specific subgroup of micronucleated cells (e.g. PIQUE).

To maintain readability, especially with the expansion of the results sections, we kept the discussion of how we envision using visual cell sorting for other MN-based applications in the discussion section.

(6) I am confused about what "cell" is referring to in much of the manuscript. Is it the nucleus + MNs only? Is it the whole cell, which one would ordinarily think it is? If so, are there additional widefield images, where one can discern cell boundaries? I found the section "MNFinder accurately ..." very hard to read and digest for this reason and other ambiguous wording. I suggest the authors take a fresh look at their manuscript and see whether the text can be improved for clarity. I did not find it an easy read overall, especially the computational part.

After re-examining how “cell” was used, we updated the text to limit its use to the MNFinder arm tasked with identifying MN-nucleus associations where the convex hull defined by these objects is used to determine the “cell” boundary. In all other cases we have replaced cell with “nucleus” because, as the reviewer points out, that is what is being analyzed and converted. We hope this is clearer.

(7) Post-FACS PPVs are not that great (Figure 3c). It depends on the question one wants to answer whether ~70% PPV is good enough. Again, would be good to comment on.

We added discussion of this result to the revision. In summary, a likely reason for the reduced PPV is that, although we maintain the cells in buffer with a Cdk1 inhibitor, we know that some proportion of the cells go through mitosis post-sorting. Since MN are frequently reincorporated into the nucleus after mitosis (Hatch et al, 2013; Zhang et al., 2015), we expect this to reduce the MN+ population. Thus, we expect that the PPV in the RNAseq population is higher than what we can measure by analyzing post-sorted cells that have been plated and analyzed later.

(8) I am thoroughly confused as to why the authors claim that their system works in the "absence of genetic perturbations" and why they emphasize the fact that their cells are non-transformed: They still needed a fluorescent label and they induce MNs with a chemical Mps1 inhibitor. (The latter is not a genetic manipulation, of course, but they still need to enrich MNs somehow. That is, their method has not been tested on a cell population in which MNs occur naturally, presumably at a very low rate, unless I missed something.) A more careful description of the benefits of their method would be good.

We apologize for the confusion on these points and hope this is clarified in the revision. We were comparing our system, which can be made using transient transfection, if desired, to current tools that disambiguate aneuploidy and MN formation by deleting parts of chromosomes or engineering double strand breaks with CRISPR to generate single chromosome-specific missegregation events. Most of these systems require transformed cancer cells to obtain high levels of recombination. In contrast, visual cell sorting can isolate micronucleated cells from any cell line that can exogenously express a protein, including primary cells and non-transformed cells like RPE1s.

Other minor points:(1) The authors should not refer to "H2B channels" but to "H2B-emiRFP703 channels". It may seem obvious to the authors but for someone reading the manuscript for the very first time, it was not. I was not sure whether there were additional imaging modalities used for H2B/nucleus/chromatin detection before I went back and read that only fluorescence images of H2B-emiRFP703 were used. To put it another way, the authors are detecting fluorescence, not histones -- unless I misunderstood something.

To address this point, we altered the text to read “H2B-emiRFP703” when discussing images of this construct. For MNFinder some images were of cells expressing H2B-GFP, which has also been clarified.

(2) If the level of zoom on my screen is such that I can comfortably read the text, I cannot see much in the figure panels. The features that I should be able to see are the size of a title. The image panels should be magnified.

In the revision, the images are appended to the end at full resolution to overcome this difficulty. Thank you for your forbearance.

**Reviewer #2 (Recommendations for the authors):**
The methods are adequately explained. The Results text narrating experiments and data analysis is clear. Interpretation of a few results could be clarified and strengthened as explained below.(1) RNAseq experiments are a good proof of principle. To strengthen their interpretation in Figures 4 and 6, I would recommend the authors cite published work on checkpoint/MPS1 loss-induced chromosome missegregation (PMID: 18545697, PMID: 33837239, PMC9559752) and consider in their discussion the 'origin' and 'proportion' of micronucleated cells and irregularly shaped nuclei expected in RPE1 lines. This will help interpret Figure 6 findings on aneuploidy signature accurately. Not being able to see an MN-specific signature could be due to the way the biological specimen is presented with a mixture of cells with 'MN only' or 'rupture' or 'MN along with misshapen nuclei'. These features may all link to aneuploidy rather than 'MN' specifically.

We appreciate the reviewer’s suggestion and added a new analysis of nuclear atypia after Mps1 inhibition in RPE1 cells to Fig. S1. Overall, we found that Mps1 inhibition significantly, but modestly, increased the proportion of misshapen nuclei and chromatin bridges. Multinucleate cells were so rare that instead of giving them their own category we included them in “misshapen nuclei.” These results are consistent with images of Msp1i treated RPE1 cells from He et al. 2019 and Santaguida et al. 2017 and distinct from the stronger changes in nuclear morphology observed after delaying mitosis by nocodazole or CENPE inhibition.

We also found that the Deep Retina segmenter used to identify nuclei in VCS MN had a significant bias against highly lobulated nuclei (Fig. S2B) that led to misshapen nuclei being largely excluded from the RNAseq analyses. As a result we found no enrichment of misshapen nuclei, chromatin bridges, or dead/mitotic nuclear morphologies in MN+ compared to MN- nuclei in our RNASeq experiments (Fig. S9A).

(2) As the authors clarify in the response letter, one round of ML is unlikely to result in fully robust software; additional rounds of ML with other markers will make the work robust. It will be useful to indicate other ML image analysis tools that have improved through such reiterations. They could use reviews on challenges and opportunities using ML approaches to support their statement. Also in the introduction, I would recommend labelling as 'rapid' instead of 'rapid and precise' method.

We updated the text to reference review articles that discuss the benefit of additional training for increasing ML accuracy and changed the text to “rapid.”

(3) The lack of live-cell studies does not allow the authors to distinguish the origin of MN (lagging chromatids or unaligned chromosomes). As explained in 1, considering these aspects in discussion would strengthen their interpretation. Live-cell studies can help reduce the dependencies on proximity maps (Figure S2).

The revised text includes new references and data (Fig. S1E) demonstrating that Mps1 inhibition strongly biases towards whole chromosome missegregation and that MN are most likely to contain a single centromere positive chromosome rather than chromatin fragments or multiple chromosomes.

(4) Mean Intersection over Union (mIOU) is a good measure to compare outcomes against ground truth. However, the mIOU is relatively low (Figure 2D) for HeLa-based functional genomics applications. It will help to discuss mIOU for other classifiers (non-MN classifiers) so that they can be used as a benchmark (this is important since the authors state in their response that they are the first to benchmark an MN classifier). There are publications for mitochondria, cell cortex, spindle, nuclei, etc. where IOU has been discussed.

We added references to classifiers for other small cellular structures. We also evaluated major sources of error in MNFinder found that false negatives are enriched in very small MN (3 to 9 pixels, or about 0.4 µm^2^ – 3 µm^2^, Fig. S6B). A similar result was obtained for VCS MN (Fig. S3B). Because small changes in the number of pixels identified in small objects can have outsized effects on mIoU scores, we suspect that this is exerting downward pressure on the mIoU value. Based on the PPV and recall values we identified, we believe that MNFinder is robust enough to use for functional genomics and screening applications with reasonable sample sizes.

(5) Figure 5 figure legend title is an overinterpretation. MN and rupture-initiated transcriptional changes could not be isolated with this technique where several other missegregation phenotypes are buried (see point 1 above).

We decided to keep the figure title legend based on our analysis of known missegregation phenotypes in Fig. S1 and S9 showing that there is no difference in major classes of nuclear atypia between MN+ and MN- populations in this analysis. Although we cannot rule out that other correlated changes exist, we believe that the title represents the most parsimonious interpretation.

Minor comments(1) The sentence in the introduction needs clarification and reference. "However, these interventions cause diverse "off-target" nuclear and cellular changes, including chromatin bridges, aneuploidy, and DNA damage." Off-target may not be the correct description since inhibiting MPS1 is expected to cause a variety of problems based on its role as a master kinase in multiple steps of the chromosome segregation process. Consider one of the references in point 1 for a detailed live-cell view of MPS1 inhibitor outcomes.

We have changed “off-target” to “additional” for clarity.

(2) In Figure 3 or S3, did the authors notice any association between the cell cycle phase and MN or rupture presence? Is this possible to consider based on FACS outcomes or nuclear shapes?

Previous work by our lab and others have shown that MN rupture frequency increases during the cell cycle (Hatch et al., 2013; Joo et al., 2023). Whether this is stochastic or regulated by the cell cycle may depend on what chromosome is in the MN (Mammel et al., 2021) and likely the cell line. Unfortunately, the H2B-emiRFP703 fluorescence in our population is too variable to identify cell cycle stage from FACS or nuclear fluorescence analysis.

(3) Figure 5 - Please explain "MA plot".

An MA plot, or log fold-change (M) versus average (A) gene expression, is a way to visualize differently expressed genes between two conditions in an RNASeq experiment and is used as an alternative to volcano plots. We chose them for our paper because most of the expression changes we observed were small and of similar significance and the MA plot spreads out the data compared to a volcano plot and allowed a better visualization of trends across the population.

(4) Page 7: "our results strongly suggest that protein expression changes in MN+ and rupture+ cells are driven mainly by increased aneuploidy rather than cellular sensing of MN formation and rupture.". This is an overstatement considering the mIOU limits of the software tool and the non-exclusive nature of MN in their samples.

We agree that we cannot rule out that an unknown masking effect is inhibiting our ability to observe small broad changes in transcription after MN formation or rupture. However, we believe we have minimized the most likely sources of masking effects, including nuclear atypia and large scale aneuploidy differences, and thus our interpretation is the most likely one.

**Reviewer #3 (Recommendations for the authors):**
Overall, the authors need to explain their methods better, define some technical terms used, and more thoroughly explain the parameters and rationale used when implementing these two protocols for identifying micronuclei; primarily as this is geared toward a more general audience that does not necessarily work with machine learning algorithms.(1) A clearer description in the methods as to how accuracy was calculated. Were micronuclei counted by hand or another method to assess accuracy?

We significantly expanded the section on how the machine learning models were trained and tested, including how sensitivity and specificity metrics were calculated, in both the results and the methods sections. The code used to compare ground truth labels to computed masks is also now included in the MNFinder module available on the lab github page.

(2) Define positive predictive value.

The text now says “the positive predictive value (PPV, the proportion of true positives, i.e. specificity) and recall (the proportion of MN found by the classifier, i.e. sensitivity)…”.

(3) Why is it a problem to use the VCS MN at higher magnifications where undersegmentation occurs? What do the authors mean by diminished performance (what metrics are they using for this?).

We have included a representative image and calculated mIoU and recall for 40x magnification images analyzed by MNFinder after rescaling in Fig. 2A. In summary, VCS MN only correctly labeled a few pixels in the MN, which was sufficient to call the adjacent nucleus “MN+” but not sufficient for other applications, such as quantifying MN area. In addition, VCS MN did much worse at identifying all the MN in 40x images with a recall, or sensitivity, metric of 0.36. We are not sure why. Developing MNFinder provided a module that was well suited to quantify MN characteristics in fixed cell images, an important use case in MN biology.

(4) The authors should compare MN that are analyzed and not analyzed using these methods and define parameters. Is there a size limitation? Closeness to the main nucleus?

We added two new figures defining what contributes to module error for both VCS MN (Fig. S3) and MNFinder (Fig. S6). For VCS MN, false negatives are enriched in very large or very small MN and tend to be dimmer and farther from the nucleus than true positives. False positives are largely misclassification of small dim objects in the image as MN. For MNFinder, the most missed class of MN are very small ones (3-9 px in area) and the majority of false positives are misclassifications of elongated nuclear blebs as MN.

(5) Are there parameters in how confluent an image must be to correctly define that the micronucleus belongs to the correct cell? The authors discussed that this was calculated based on predicted distance. However, many factors might affect proper calling on MN. And the authors should test this by staining for a cytosolic marker and calculating accuracy.

We updated the text with more information about how the cytoplasm was defined using leaky 2x-Dendra2-NLS signal to analyze the accuracy of MN/nucleus associations (Fig. S2G-H). In addition, we quantified cell confluency and distance to the first and second nearest neighbor for each MN in our training and testing image datasets. We found that, as anticipated, cells were imaged at subconfluent concentrations with most fields having a confluency around 30% cell coverage (Fig. S2E) and that the average difference in distance between the closest nucleus to an MN and the next closest nucleus was 3.3 fold (Fig. S2F). We edited the discussion section to state that the ability of MN/nuclear proximity to predict associations at high cell confluencies would have to be experimentally validated.

(6) The authors measure the ratio of Dendra2(Red) v. Dendra2 (Green) in Figure 3B to demonstrate that photoconversion is stable. This measurement, to me, is confusing, as in the end, the authors need to show that they have a robust conversion signal and are able to isolate these data. The authors should directly demonstrate that the Red signal remains by analyzing the percent of the Red signal compared to time point 0 for individual cells.

We found a bulk analysis to be more powerful than trying to reidentify individual cells due to how much RPE1 cells move during the 4 and 8 hours between image acquisitions. In addition, we sort on the ratio between red and green fluorescence per cell, rather than the absolute fluorescence, to compensate for variation in 2xDendra-NLS protein expression between cells. Therefore, demonstrating that distinct ratios remained present throughout the time course is the most relevant to the downstream analysis.

To address the reviewer’s concern, we replotted the data in Fig. 3B to highlight changes over time in the raw levels of red and green Dendra fluorescence (Fig. S7D). As expected, we see an overall decrease in red fluorescence intensity, and complementary increase in green fluorescence intensity, over 8 hours, likely due to protein turnover. We also observe an increase in the number of nuclei lacking red fluorescence. This is expected since the well was only partially converted and we expect significant numbers of unconverted cells to move into the field between the first image and the 8 hour image.

(7) The authors isolate and subsequently use RNA-sequencing to identify changes between Mps1i and DMSO-treated cells. One concern is that even with the less stringent cut-off of 1.5 fold there is a very small change between DMSO and MPS1i treated cells, with only 63 genes changing, none of which were affected above a 2-fold change. The authors should carefully address this, including why their dataset sees changes in many more pathways than in the He et al. and Santaguida et al. studies. Is this due to just having a decreased cut-off?

The reviewer correctly points out that we observed an overall reduction in the strength of gene expression changes between our dataset of DMSO versus Mps1i treated RPE1 cells compared to similar studies. We suggest a couple reasons for this. One is that the log_2_ fold changes observed in the other studies are not huge and vary between 2.5 and -3.8 for He et al., 3.3 and -2.3 for Santaguida et al., and -0.8 and 1.6 for our study. This variability is within a reasonable range for different experimental conditions and library prep protocols. A second is that our protocol minimizes a potential source of transcriptional change – nuclear lobulation – that is present in the other datasets.

For the pathway analysis we did not use a fold-change cut-off for any data set, instead opting to include all the genes found to be significantly different between control and Mps1i treated cells for all three studies. Our read-depth was higher than that of the two published experiments, which could contribute to an increased DEG number. However, we hypothesize that our identification of a broader number of altered pathways most likely arises from increased sensitivity due to the loss of covering signal from transcriptional changes associated with increased nuclear atypia. Additional visual cell sorting experiments sorting on misshapen nuclei instead of MN would allow us to determine the accuracy of this hypothesis.

(8) Moreover, clustering (in Figure 5E) of the replicates is a bit worrisome as the variances are large and therefore it is unclear if, with such large variance and low screening depth, one can really make such a strong conclusion that there are no changes. The authors should prove that their conclusion that rupture does not lead to large transcriptional changes, is not due to the limitations of their experimental design.

We agree with the reviewers that additional rounds of RNAseq would improve the accuracy of our transcriptomic analysis and could uncover additional DEGs. However, we believe the overall conclusion to be correct based on the results of our attempt to validate changes in gene expression by immunofluorescence. We analyzed two of the most highly upregulated genes in the ruptured MN dataset, ATF3 and EGR1. Although we saw a statistically significant increase in ATF3 intensity between cells without MN and those with ruptured MN, the fold change was so small compared to our positive control (100x less) that we believe it is it is more consistent with a small increase in the probability of aneuploidy rather than a specific signature of MN rupture.

(9) The authors also need to address the fact that they are using RPE-1 cells more clearly and that the lack of effect in transcriptional changes may be simply due to the loss of cGAS-STING pathway (Mackenzie et al., 2017; Harding et al., 2017; etc.).

As we discuss above in the public comments section, the literature is clear that MN do not activate cGAS in the first cell cycle after their formation, even upon rupture. Therefore, we do not expect any changes in our results when applied to cGAS-competent cells. However, this expectation needs to be experimentally validated, which we plan to address in upcoming work.